# Sensory input drives rapid homeostatic scaling of the axon initial segment in mouse barrel cortex

Nora Jamann [1,2,3], Dominik Dannehl[3], Nadja Lehmann [3], Robin Wagener[4], Corinna Thielemann[3], Christian Schultz[3], Jochen Staiger[5], Maarten H. P. Kole [1,2✉] & Maren Engelhardt[3✉]

The axon initial segment (AIS) is a critical microdomain for action potential initiation and implicated in the regulation of neuronal excitability during activity-dependent plasticity. While structural AIS plasticity has been suggested to fine-tune neuronal activity when network states change, whether it acts in vivo as a homeostatic regulatory mechanism in behaviorally relevant contexts remains poorly understood. Using the mouse whisker-to-barrel pathway as a model system in combination with immunofluorescence, confocal analysis and electro-physiological recordings, we observed bidirectional AIS plasticity in cortical pyramidal neurons. Furthermore, we find that structural and functional AIS remodeling occurs in distinct temporal domains: Long-term sensory deprivation elicits an AIS length increase, accompanied with an increase in neuronal excitability, while sensory enrichment results in a rapid AIS shortening, accompanied by a decrease in action potential generation. Our findings highlight a central role of the AIS in the homeostatic regulation of neuronal input-output relations.

[1] Axonal Signaling Group, Netherlands Institute for Neurosciences (NIN), Royal Netherlands Academy for Arts and Sciences (KNAW), Amsterdam, The Netherlands. [2] Cell Biology, Neurobiology and Biophysics, Department of Biology, Faculty of Science, Utrecht University, Utrecht, The Netherlands. [3] Institute of Neuroanatomy, Mannheim Center for Translational Neuroscience (MCTN), Medical Faculty Mannheim, Heidelberg University, Mannheim, Germany. [4] Clinic of Neurology, University Hospital Heidelberg, Heidelberg, Germany. [5] Institute of Neuroanatomy, University Medical Center, Georg August University of Göttingen, Göttingen, Germany. ✉email: m.kole@nin.knaw.nl; maren.engelhardt@medma.uni-heidelberg.de

The axon initial segment (AIS) serves as the cellular substrate for action potential (AP) initiation in most neurons of the central nervous system[1–3]. It is characterized by a periodic nanoscale arrangement of actin and scaffolding proteins, including Ankyrin-G (ankG) and βIV-spectrin[4,5], which tether voltage-gated ion channels, extracellular membrane proteins and receptors at the axonal membrane (reviewed in[6]). In particular the clustering of sodium and potassium channels at the AIS plays an important role in giving rise to AP initiation[7–9].

Recent work has shown that the AIS dynamically adapts its length and/or location within the axon relative to the soma, thereby modulating neuronal input/output properties ([10–15], reviewed in[16,17]). Depending on the excitation state of the neuronal network both in vitro and in vivo, individual neurons can undergo structural and/or functional AIS plasticity, including modulation of channel density or composition, to adapt to changes in synaptic drive ([18–21], reviewed in[16,22]). For example, triggering chronic depolarization in neuronal networks in vitro[10,12] or using a complete and irrevocable elimination of sensory input in vivo[11] leads to significant geometrical and functional AIS changes. This activity-dependent AIS plasticity is, in part, regulated by calcium ($Ca^{2+}$)-dependent pathways[12,23]. These studies are also in good support of AIS remodeling observed in disease or injury models ([24–26], reviewed in[27]). In addition to the up- or downregulation of excitability by AIS changes, cell-type-specific remodeling of the AIS has also been found to occur either via long-term (days to weeks;[11,28–31]) or short-term manipulations of just a few hours[12,20,21].

Structural adaptations in the AIS have often been hypothesized to reflect a homeostatic mechanism at the network level to ensure a dynamic equilibrium of neuronal activity (reviewed in[32]). However, such a role of the AIS in vivo remains to be demonstrated. To test whether AIS plasticity occurs in a behaviorally relevant context, we utilized the mouse whisker-to-barrel system[33]. This pathway conveys tactile information from vibrissae on the rodent whisker pad to the barrel field in primary somatosensory cortex (S1BF) in a strictly organized hierarchy and is one of the best-studied sensory systems (reviewed in[34]). Thalamic input is largely received by layer IV spiny stellate neurons and propagated to supragranular (layer II/III) neurons. In addition, infragranular (layer V) principal neurons receive direct thalamic input[35]. The S1BF undergoes activity-dependent maturation with several critical periods during postnatal development (reviewed in[16,36]), ultimately resulting in a precise somatotopic representation of each whisker in a cortical barrel layer IV[33].

Here, we employed a range of sensory deprivation and enrichment paradigms in behaving mice. We investigated whether the AIS geometry and AP generation were affected following these modifications of sensory input, and which time-course these changes followed. We found that AIS plasticity follows two distinct temporal patterns: deprivation of the whisker-to-barrel pathway for 15 days or longer caused long-term AIS elongation in S1BF layer II/III neurons, accompanied by an increase in pyramidal neuron excitability. In contrast, increasing the activity of the same neuronal population for just 1–3 hours (h) via exposure to an enriched environment resulted in AIS shortening and increased the threshold for AP generation, reducing pyramidal neuron output. In summary, our results indicate a temporally diverse, bidirectional, activity-dependent remodeling of the AIS and input-output properties, supporting its role for homeostatic adaptation under physiological conditions in vivo.

## Results

### The AIS undergoes periods of structural plasticity during development. To examine the time course of AIS development in

the whisker-to-barrel system, we first investigated normal AIS maturation at selected time points during S1BF development of mice. Based on immunofluorescent staining, AIS length was quantified from the late embryonic period (E20) throughout postnatal development (P1, P3, 7, 13, 15, 21, 28, 35) into adulthood (P45, P180; each time point $n = 6$ animals, at least 100 AIS per animal in layers II/III and V of S1BF; Fig. 1a, b). For E20, superficial and deep layers were not clearly distinguishable yet, therefore data were pooled (Fig. 1a, b). Antibodies against the AIS scaffolding proteins βIV-spectrin and ankG, well-established targets for morphometrical analysis of the AIS[28–30], were utilized. Double stainings revealed an overlap of immunofluorescent signals along the AIS for both proteins (Fig. S1a). Therefore, both served as length and distance markers in this study. While we did not employ specific markers for neuron subtypes, the interneuron population in S1BF is relatively scarce, as illustrated by a representative panel from the cortex of a parvalbumin-tdTomato reporter mouse line (Fig. S1b). Additionally, AIS of interneurons often appear clearly thinner without the tapering in diameter after the axon hillock that is typical for pyramidal neurons (Fig. S1c, d;[37,38]). Also, they frequently run horizontally or even towards the cortical surface and could therefore be excluded from analysis (Fig. S2c, d). Consequently, we postulate that the majority of AIS analyzed in this study are from cortical pyramidal neurons.

Immunoblot analysis was directed against the known isoforms of ankG, the main regulator of AIS assembly and maintenance[39]. In line with previous reports[28,40], we found that axon initial segments (AIS) elongated during the early postnatal period until the end of the second postnatal week, a time at which mice begin active whisking to explore their environment[41], indicated by gray boxes in Fig. 1b. With the onset of active whisking behavior, however, a significant shortening of AIS length was observed both in pyramidal neurons of layers II/III and V (Fig. 1b). Adult animals then maintained an intermediate AIS length until P45, after which no more significant length changes were observed (Fig. 1b). Although the overall developmental profiles of layers II/III and V were comparable, AIS length in layer V started to decrease after P10 (Fig. 1b). AIS in layer V were generally shorter than in layer II/III (average adult S1BF layer II/III: ~22 μm, layer V: ~19 μm) in accordance with previously published data from visual and somatosensory cortex[28,31].

Previous studies have shown that the giant (480 kDa) isoform of *Ank3*, which encodes ankG, serves as the major organizer of AIS assembly[39]. To test whether the developmental increase in AIS length is accompanied by an upregulation of the ankG isoforms (190, 270 and 480 kDa), we performed immunoblot analysis (Fig. 1c, d). The results showed that consistent with the AIS elongation during the early postnatal period (P1–P15), there was a developmental increase in all ankG isoforms (Fig. 1d). Interestingly, the expression of the 190 kDa ankG isoform further increased until adulthood, which may be in part explained by synaptogenesis and targeting of this isoform to postsynaptic structures[42].

### AIS elongation is triggered by long-term sensory deprivation.
The significant reduction in AIS length seen from P15 (in layer II/III) or P10 (in layer V) and onwards coincides with the onset of explorative, active whisking behavior in mice[41]. A similar AIS maturation pattern (early postnatal elongation, followed by length reduction and subsequent stable adult length) was previously described for the primary visual cortex and shown to be regulated by the onset of vision at P13–14 (Ref. [28]). We therefore hypothesized that the structural remodeling of the AIS occurs due to developmental changes in tactile input when mice begin to actively explore their environment. We tested this hypothesis by

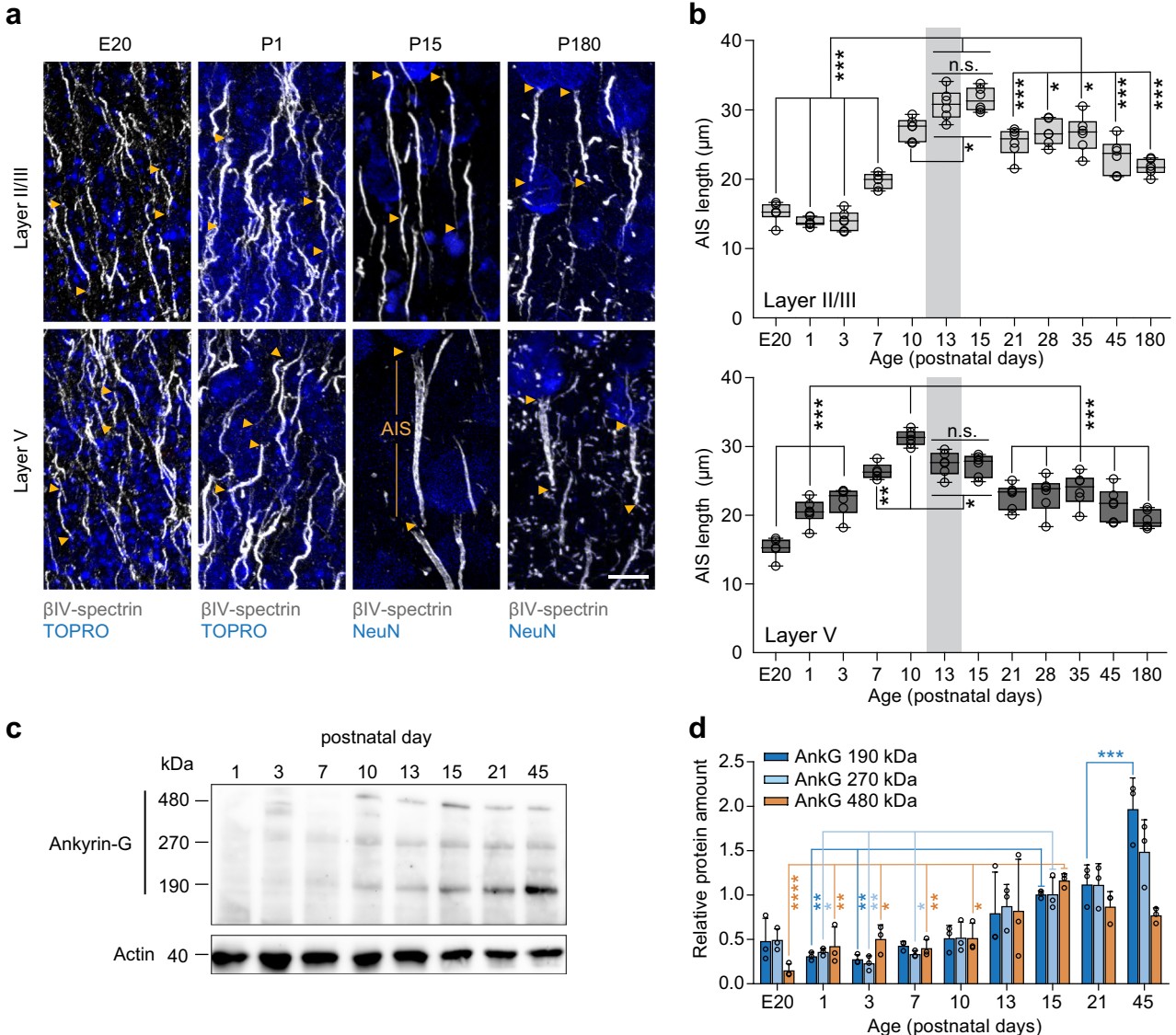

**Fig. 1 AIS development in S1BF in vivo. a** Representative confocal images of AIS length maturation for one embryonal and three postnatal ages (E20, P1, P15, and P180) in cortical layers II/III and V. Immunostaining against ßIV-spectrin (gray), TOPRO (blue) or NeuN (blue) as indicated. Note the long AIS at P15 in both layers and the appearance of nodes of Ranvier in P180 animals. Arrowheads indicate start and end of AIS; orange lines indicate extent of a single AIS at P15. Scale bar 10 μm. **b** Population data of AIS lengths from E20 to P180 in layer II/III (top) and V (bottom). E20 data was pooled for supra- and infragranular layers. Initially, AIS length increases until P13–15, after which it decreases. Gray bars indicate the onset of active exploration and whisking (P12–14). Adult animals maintain an intermediate AIS length throughout life (for layer II/III and layer V, One-way ANOVA ***$P < 0.001$, Holm–Sidak's post-hoc comparisons, $n = 6$ biologically independent experiments per age group, $n = 5$ biologically independent experiments for P7; >100 AIS per animal. For all comparisons, *$P < 0.05$, **$P < 0.01$, ***$P < 0.001$, summary of $P$ values see Tables S1 and S2). Boxplots: median with 25–75% interval, error bars show minimum to maximum values. **c** Representative immunoblot of the major ankG isoforms (190, 270, 480 kilodalton, kDa) for postnatal developmental stages. Actin was used as a loading control. **d** Quantification of immunoblot data derived from three independent experiments run in parallel. Protein expression of the main ankG 480 kDa isoform peaks at P15 (Two-way ANOVA ***$P < 0.0001$ for time, *$P = 0.067$ for isoform, ***$P = 0.0007$ (time x isoform) for the interaction, $n = 3$ biologically independent experiments per age group, Tukey's multiple comparisons tests *$P < 0.05$, **$P < 0.01$, ***$P < 0.001$, only a selection of statistical comparisons are depicted for visualization. Summary of all $P$ values in Table S3). Data shown as mean ± SD.

conducting sensory deprivation experiments via daily, bilateral whisker trimming from birth (P0) to P15, P21, and P45, respectively (Group 1, Fig. 2a–c). P15 coincides with the closure of the critical period for layer IV to II/III synapses[36,43], while P21 was identified as the time point of significant AIS length reduction during development (Fig. 1b). At the P15, P21, and P45 endpoints, mice were perfused and S1BF cryosections examined via immunofluorescence and morphometrical analysis of AIS length in layer II/III (Fig. 2b) and layer V (Fig. S2a) neurons. Bilateral whisker trimming resulted in a significant lengthening of AIS at all time

points in Group 1 in layers II/III (Two-way ANOVA $P < 0.0001$ (deprivation), Fig. 2b, c), but not in layer V (measured at P15 and P21, Two-way ANOVA $P = 0.20$ (deprivation), Fig. S2a). In a second group, we applied a shorter trimming period of 5 days (P10–P15). However, this did not lead to AIS length changes in layer II/III (Two-way ANOVA $P = 0.33$ (deprivation) Fig. 2a, d), indicating a minimum of 2 weeks of deprivation to induce significant length changes during development. Length frequency distribution analysis of the AIS in Group 1 showed that with whisker trimming until P15, the number of longer AIS significantly

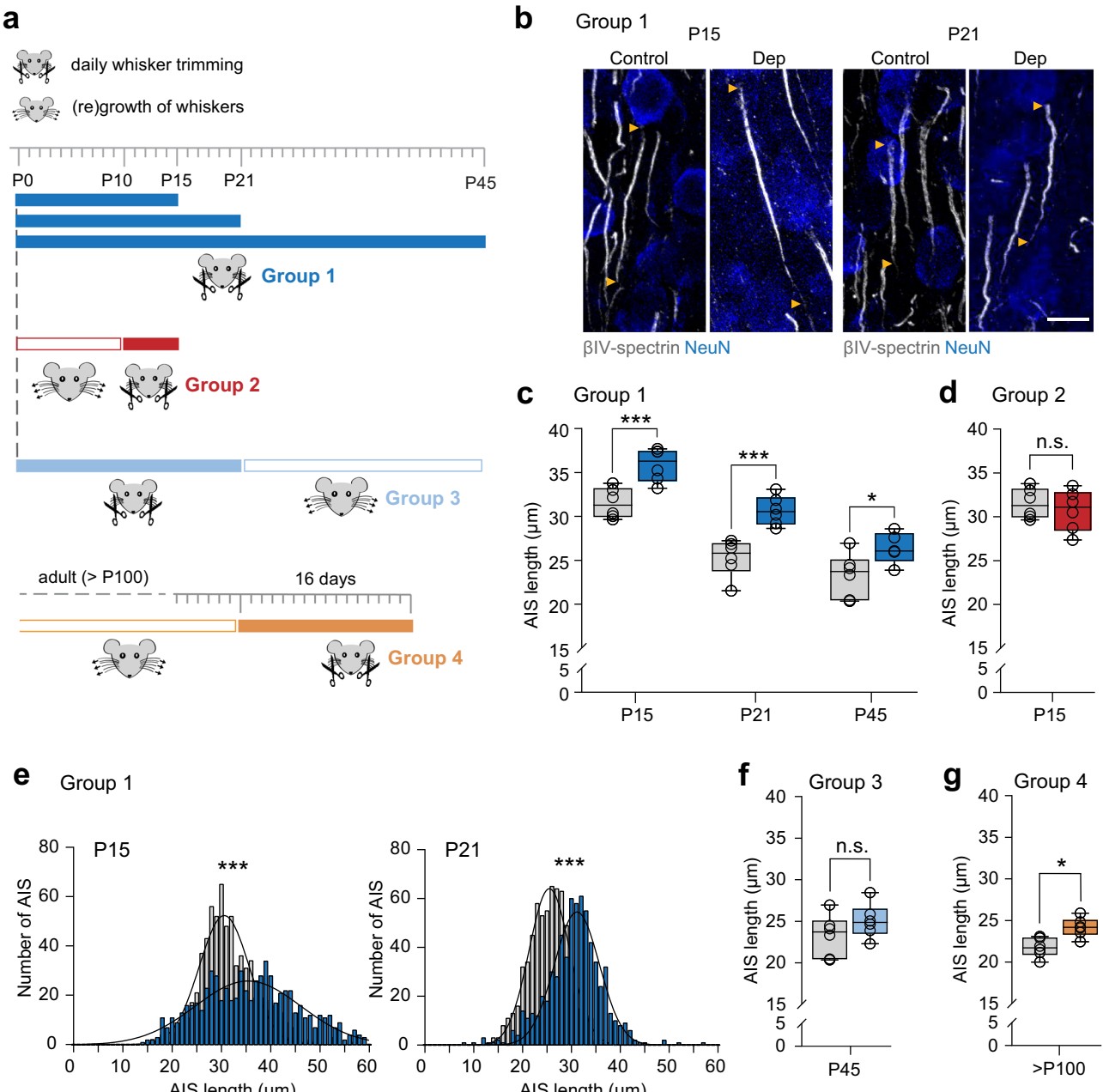

**Fig. 2 Long-term sensory deprivation elongates the AIS in layer II/III of young and adult mice. a** Experimental designs for whisker trimming. Group 1, daily bilateral trimming from P0 to P15, P21 or P45, respectively. Group 2, daily bilateral trimming from P10 to P15. Group 3, daily bilateral trimming until P21, followed by regrowth until P45. Group 4, daily bilateral trimming for 16 days in mice > P100. **b** Representative confocal images of layer II/III AIS stained for ßIV-spectrin (gray) and NeuN (blue) in control and deprived (Dep) mice. Arrowheads indicate start and end of AIS. Scale bar 10 µm. **c** Population data for Group 1 (control = gray; deprived = dark blue). Deprivation leads to longer AIS (Two-way ANOVA $P < 0.0001$ for deprivation and development, $P = 0.417$ (deprivation x development), >100 AIS/animal, $n = 5$ biologically independent experiments for P45 Dep, $n = 6$ biologically independent experiments for all other groups, Sidak's multiple comparison tests $*P = 0.042$, $**P = 0.002$, $***P = 0.0002$). **d** Population data for Group 2 (control = gray; deprived = red). Deprivation from P10 to P15 did not lead to significant length changes (unpaired two-sided $t$-test $P = 0.53$, >100 AIS/ animal, $n = 6$ biologically independent experiments). **e** Length frequency histograms for P15 and P21 (control = gray; deprived = dark blue) Dep vs Ctrl, >600 AIS per condition, Kolmogorov–Smirnov test $***P < 0.0001$, median: P15: Ctrl 30.8 µm, Dep 35.8 µm, P21 Ctrl 25.6 µm, Dep 31.0 µm). **f** Population data for Group 3 (deprived = light blue; age-matched controls = gray) at P45. Upon whisker regrowth, AIS length returned to control levels (unpaired two-sided $t$-test $P = 0.22$, >100 AIS/animal, $n = 6$ biologically independent experiments). **g** Population data for Group 4 (adult > P100; deprived = orange; age-matched controls = gray). Deprived animals showed significantly longer AIS after two weeks of whisker trimming (unpaired two-sided $t$-test $**P = 0.0046$, >100 AIS/animal, $n = 6$ biologically independent experiments). Boxplots: median with 25–75% interval, error bars show minimum to maximum values.

increased (Kolmogorov–Smirnov test $P < 0.0001$), with some AIS reaching even 60 μm (Fig. 2e). Therefore, the width of the curve was more than twice as wide in the deprived group (full width at half maximum (FWHM) P15 Ctrl 11.9 μm vs P15 Dep 25.3 μm). By contrast, trimming until P21 led to a significant shift in the length median (Kolmogorov–Smirnov test $P < 0.0001$), but the full width at half maximum remained similar to control AIS (FWHM P21 Ctrl 9.79 μm vs P21 Dep 10.86 μm, respectively, Fig. 2e).

To test whether AIS length change is reversible after sensory deprivation, another group of mice (Group 3) was trimmed daily from P0 to P21 and sacrificed at P45, after whiskers had regrown (Fig. 2a, f). The results showed that in this group, AIS length returned to mature levels (unpaired $t$-test $P = 0.22$, Fig. 2f). Again, we observed no length differences in layer V neurons (unpaired $t$-test $P = 0.88$, Fig. S2b). To control for any influence of sensory stimulation by the handling or trimming itself, we performed additional control experiments in which animals were handled daily and whiskers were ruffled, but not trimmed. In these animals, no changes in AIS length were observed (unpaired $t$-test $P = 0.16$, Fig. S2c). Finally, to test whether AIS plasticity would also occur in adult mice, we applied whisker trimming after P100 for 16 days (Group 4). Strikingly, a significant AIS elongation was still observed in layer II/III (unpaired $t$-test, $P = 0.0046$, Fig. 2g), but not in layer V (unpaired $t$-test $P = 0.95$, Fig. S2d).

In summary, these experiments indicate that an AIS length increase is induced by long-term sensory deprivation in the whisker-to-barrel system during development and adulthood, which can be reversed by restoration of tactile input.

**Long-term deprivation leads to changes in neuronal excitability**. Based on theoretical and experimental work, a longer AIS should correspond to increased neuronal excitability by reducing the threshold for AP generation[11,17]. In order to examine whether sensory-deprivation induced AIS length plasticity was associated with electrophysiological changes, we performed whole-cell patch-clamp recordings in control and deprived mice, trimmed daily and bilaterally from P0 to P15. At P15 (±2 days), acute slices were prepared and active and passive membrane properties were recorded from layer II/III pyramidal neurons (Dep, $n = 18$ cells from 7 mice; Ctrl, $n = 15$ cells from 7 mice). The data showed that resting membrane properties were not changed between the deprivation and control group (resting membrane potential (RMP): Ctrl −81.76 ± 3.79 mV, Dep −78.62 ± 5.13 mV, $P = 0.056$, input resistance ($R_N$): Ctrl 303.12 ± 73.98 MΩ, 347.45 ± 72.02 MΩ, $P = 0.092$). To assess AP threshold and firing properties, we injected depolarizing current steps (Fig. 3a, c). The results indicated that deprivation significantly increased firing rates just above threshold (Two-way ANOVA $P < 0.0001$ for the factor deprivation, Holm–Sidak's multiple comparisons, for 100 pA, $P = 0.0037$ and for 150 pA, $P = 0.014$, Fig. 3a). However, deprivation did not affect the maximum firing rates (~20 Hz, Fig. 3a). Furthermore, the current at the maximum slope was significantly reduced in the deprivation group (unpaired $t$-test $P = 0.028$, Fig. 3b). Since the resting membrane properties were not different between the groups, these results suggest that the threshold for APs may be reduced. Using 20 ms duration step current injections, we quantified threshold properties for AP generation and found that deprivation led to a ~50 pA reduction in current threshold, without changing the voltage threshold (unpaired $t$-tests, $P = 0.015$ and $P = 0.070$, respectively; Fig. 3c, d). Furthermore, neither AP half-width, amplitude nor phase-plane properties of the AP were different between the two groups (Fig. S3a, b). The reduced current threshold for AP generation is consistent with the increased AIS length identified in the

population analysis (Fig. 2). In order to test such a link more directly, we performed correlation analyses between the current threshold and the specific AIS length (Fig. 3e, f) as well as the AIS onset position relative to the soma (Fig. S3c), by immunostaining post-hoc against βIV-spectrin and the biocytin fill of the recorded neuron (Fig. 3e). In agreement with our hypothesis, these data revealed a significant correlation ($r^2 = 0.36$, $P = 0.029$) between AIS length and current threshold (Fig. 3f). The AIS onset position however did not significantly correlate with current threshold ($r^2 = 0.0001$, $P = 0.97$, Fig. S3c). Finally, to examine the network activity we characterized synaptic input onto layer II/III neurons by measuring in voltage-clamp the spontaneous excitatory post-synaptic currents (PSCs) at −90 mV (Fig. 3g). The deprived cells received PSCs at a higher frequency, albeit with on average a lower amplitude (Fig. 3g, h, unpaired $t$-test $P = 0.0042$ and $P = 0.035$, respectively).

Taken together, these data show that a long-term reduction of sensory tactile input increases the intrinsic excitability of layer II/III pyramidal neurons and correlates with increased AIS length.

**AIS undergo rapid structural plasticity after induced exploratory activity in vivo**. Our data so far suggest that in vivo, an increase in AIS length requires long periods of sensory deprivation (weeks, Fig. 2). In contrast, studies applying depolarizing conditions in vitro showed that AIS length changes can be triggered more rapidly, within minutes to hours of increased in vitro network activity[12,19]. We therefore tested whether rapid AIS plasticity could be evoked in vivo in a behaviorally relevant context, by placing young adult wild type mice at P28 in an enriched environment (EE) and exposing whiskers to a larger range of novel stimuli. For EE, we placed a variety of novel objects and different types of bedding in a large home-cage, enabling enhanced explorative behavior[44]. 12 h prior to the experiment, mice were subjected to unilateral whisker trimming (Fig. 4a). Consequently, the S1BF contralateral to the intact whisker pad received increased sensory input via the whisker-to-barrel pathway (EE and whiskers intact side) compared to the ipsilateral side (Ctrl side)[45]. To exclude possible short-term deprivation effects of whisker trimming on AIS length, we included a "0 h" group, where animals that received unilateral whisker trimming were sacrificed immediately after the 12 h post whisker trimming without being placed in the EE cage (Fig. 4a). Experimental groups remained in EE for 1, 3 and 6 h, respectively. At each end point, mice were sacrificed and S1BF tissue blocks were processed for morphometrical analysis of AIS parameters ($n = 6$ per group, $n = 5$ for 1 h group, at least 100 AIS per animal, Fig. 4).

Increased neuronal activity selective to the EE side of S1BF was confirmed by immunofluorescent detection of the immediate early gene product c-Fos (Fig. 4b), which is upregulated within very short time frames after an increase in neuronal activity occurs[44,46]. At the 0 h time point, only very scarce immunosignal was detectable (<1% c-Fos–positive (c-Fos$^+$) of all analyzed AIS-endowed cells in layer II/III of S1BF; Fig. 4b, c). In contrast, 1 h after EE exposure, layer II/III neurons showed a significant upregulation in c-Fos expression (~19% c-Fos$^+$; Fig. 4b, c), while control cells remained mostly negative for c-Fos (~2% c-Fos$^+$, Fig. 4b, c). C-Fos was further increased in expression after 3 h (~38% EE vs. ~6% Ctrl, Fig. 4b, c) and decreased to baseline levels after 6 h of EE exposure (Fig. 4b, c).

Next, we asked whether these rapid changes in S1BF network activity after EE triggered structural AIS remodeling in layer II/III cortical neurons. Population analysis of mean AIS length in layer II/III in the EE and Ctrl hemispheres revealed an AIS length reduction at 1 h after EE exposure (Two-way RM ANOVA, Sidak's multiple comparisons test $P = 0.043$, Fig. 4d). This effect

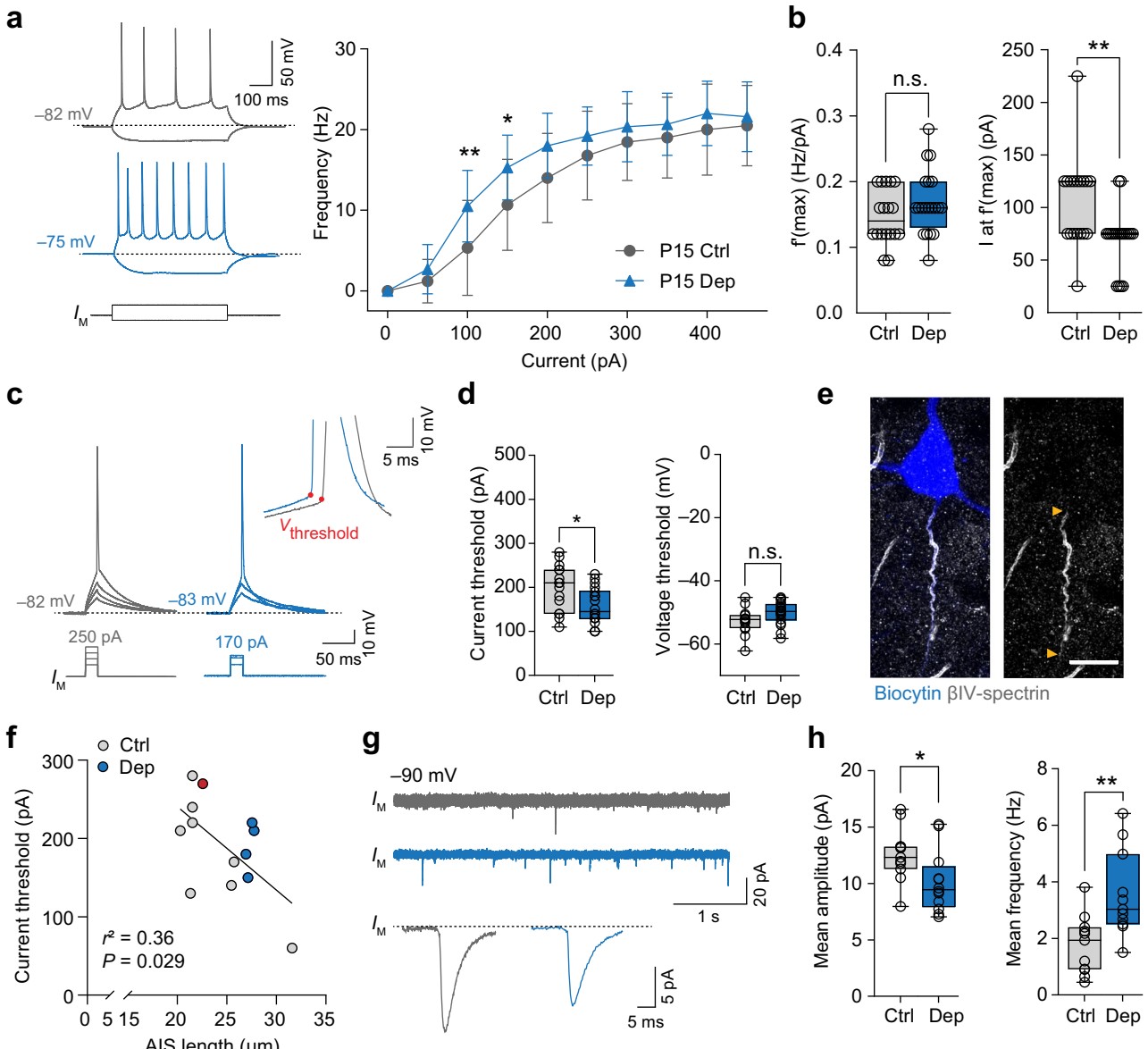

**Fig. 3 Whisker trimming increases neuronal excitability in layer II/III pyramidal neurons. a** Left: Representative traces of AP trains elicited by current injection (500 ms, −50 pA, +100 pA). Note the increased firing frequency in the deprived neurons (blue trace). Right: Input/frequency relationship as determined by 500 ms injections of increasing currents. The deprivation group showed significantly increased firing frequencies (Two-way ANOVA ***$P <$ 0.0001 for current injection and deprivation, Holm–Sidak's multiple comparison test *$P = 0.014$, **$P = 0.0037$, $n = 15$ Ctrl cells, 20 Dep cells from at least 6 biologically independent experiments). Data are presented as mean ± SD. **b** Analysis of maximum slope f'(max) of the $I$-$f$ curve and current at the maximum slope $I$ at f'(max). Current at f'(max) was significantly lower in the deprivation group (Mann–Whitney test **$P = 0.0028$). Maximum slope was unchanged (unpaired two-sided $t$-test $P = 0.15$; $n = 14$ Ctrl cells, 18 Dep cells from at least 6 biologically independent experiments). **c** Representative traces of single APs elicited by 20 ms current injections; 10 pA increment injections to determine current and voltage threshold (inset). **d** Deprived neurons had significantly lower current threshold; voltage threshold was unchanged (unpaired two-sided $t$-test, current threshold *$P = 0.015$, voltage threshold $P = 0.070$ $n = 15$ Ctrl cells, 18 Dep cells from at least 6 biologically independent experiments). **e** Representative confocal image of a P15 Dep neuron filled with biocytin (blue) for post-hoc determination of AIS length (βIV-spectrin, gray). Arrowheads indicate start and end of AIS. Scale bar 10 μm. **f** Correlation analysis of the relationship between AIS length and current threshold. Results of linear regression analysis indicated in figure ($n = 13$ cells from at least 6 biologically independent experiments). Red dot indicates sample cell from image in **e**. **g** Left: Representative traces of spontaneous postsynaptic currents (PSCs) recorded at −90 mV. Top trace: 6 s of recording. Bottom trace: averaged PSCs across the entire recording session of 2 min for two sample cells. Right: Mean amplitude and frequency of PSCs. Deprived neurons received significantly more input, with on average a lower amplitude (unpaired two-sided $t$-test, amplitude: *$P = 0.035$, frequency: **$P = 0.0042$, $n = 11$ cells for Ctrl and Dep from at least 6 biologically independent experiments). Boxplots: median with 25–75% interval, error bars show minimum to maximum values.

was significantly increased 3 h after EE, leading to a shortening of on average ~3 μm (Two-way RM ANOVA, Sidak's multiple comparisons test $P = 0.0051$, Fig. 4d). AIS length subsequently returned to control levels after 6 h of EE (Fig. 4d). No significant

AIS length alterations were detected in layer V (Two-way RM ANOVA $P > 0.05$ for time, EE and time × EE, Fig. S4a).

To test whether the AIS length reversal after 6 h was due to habituation to the novel environment, we examined a group of mice

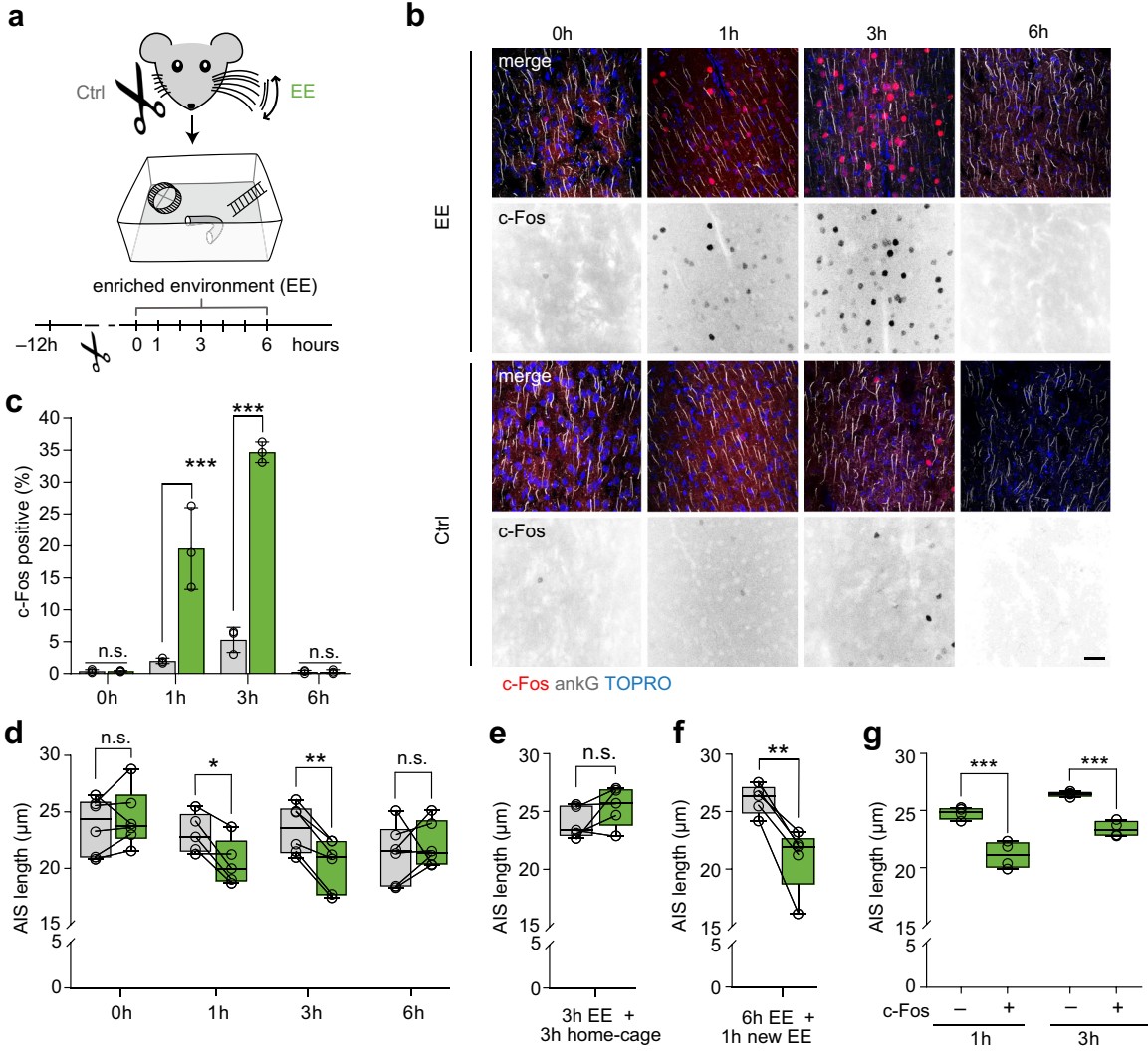

**Fig. 4 Rapid structural AIS remodeling after exposure to an enriched environment (EE). a** Experimental design: P28 mice ($n = 5$–6 biologically independent experiments/time point), exposed to EE conditions 12 h after unilateral whisker trimming. Mice remained in EE for 0, 1, 3, and 6 h, respectively, and were sacrificed immediately after. Layer II/III S1BF pyramidal neurons contralateral to the remaining whiskers represented stimulated side (EE); ipsilateral S1BF served as control (Ctrl). **b** Representative confocal images of EE and Ctrl after immunostaining of layer II/III neurons at 0, 1, 3, and 6 h EE exposure. AIS demarked by ankG (grey), with TOPRO (blue). Staining against c-Fos (red) served as indicator of neuronal activity. Inverted black & white panels highlight the increasing c-Fos signal over time. Scale bar 30 µm. **c** Quantification of c-Fos expression indicates rapid and significant upregulation after 1 h of EE (19.6% c-Fos+ neurons) with a peak expression after 3 h EE (38% c-Fos+ neurons), and downregulation after 6 h EE (1% c-Fos+ neurons). Two-way ANOVA ***$P < 0.0001$ for time, EE and EE x time, Sidak's multiple comparisons test ***$P < 0.0001$, $n = 3$ biologically independent experiments. Data are presented as mean ± SD. **d** Length reduction in S1BF layer II/III neurons (1 h, 3 h EE). AIS length reduction is reversed after 6 h EE (Two-way RM ANOVA $P = 0.021$ for the factor EE, $P = 0.16$ for the factor time (ns), **$P = 0.0046$ for EE x time, Sidak's multiple comparisons test *$P = 0.043$, **$P = 0.0051$, $n = 5$–6 biologically independent experiments; > 100 AIS/animal). Lines indicate matching data from the two hemispheres of individual mice. **e** Reversal of EE effects: After 3 h EE, AIS length normalized in home-cage (paired two-sided $t$-test $P = 0.16$, $n = 5$ biologically independent experiments, > 100 AIS/animal). **f** Exposure to a new EE induced AIS length shortening (paired two-sided $t$-test **$P = 0.0035$, $n = 5$ biologically independent experiments, >100 AIS/animal). **g** Population analysis (c-Fos+ and c-Fos− cells) after EE reveals a significantly decreased AIS length in c-Fos+ neurons (One-way ANOVA $P < 0.0001$, Sidak's multiple comparisons test ***$P < 0.0001$). Boxplots: median with 25–75% interval, error bars show minimum to maximum values.

which were placed in EE for 3 h and subsequently placed back into their home-cage environment for another 3 h. The AIS length in this group similarly remained unchanged in comparison to controls (paired $t$-test, $P = 0.162$, Fig. 4e), suggesting a rapid reversibility when sensory input is decreased to home-cage environment levels. Furthermore, we tested whether the sensory stimulation induced AIS length shortening can re-occur by exposure to a novel EE after 6 h exposure to the first EE. The results showed that 1 h of exposure to the new EE was sufficient to trigger a significant AIS length shortening (paired $t$-test $P = 0.0035$, Fig. 4f).

Next, we asked whether the AIS shortening we observed in layer II/III is global or selective to neurons activated by the EE exposure (c-Fos+ neurons). AIS length of c-Fos+ neurons was analyzed by immunostaining against ankG in combination with c-Fos and detection of nuclei via TOPRO in order to identify c-Fos negative (c-Fos−) neurons and their corresponding AIS. Consistent with the role of excitability, we found that AIS were significantly shorter (~3 µm) in c-Fos+ neurons compared to surrounding c-Fos− neurons both 1 and 3 h post EE exposure on the EE hemisphere (One-way ANOVA $P < 0.0001$, Fig. 4g). In

further support, we performed a similar analysis on the Ctrl hemisphere at 3 h, where a small fraction of neurons were c-Fos[+] and found the same effect, with shorter AIS in the c-Fos[+] population (unpaired t-test $P = 0.0046$, Fig. S4b).

Taken together, a 1 h exposure to an enriched environment suffices to trigger AIS shortening, requiring an increased sensory input via the whisker pathway.

**Rapid structural AIS remodeling triggers changes in neuronal excitation.** What are the functional consequences of the observed rapid structural AIS changes in pyramidal neurons of S1BF layer II/III? To address this question, we performed whole-cell patch-clamp recordings in acute slices in mice that were exposed to 3 h of EE as outlined in Fig. 4a. The passive properties were unchanged between the two groups (RMP: Ctrl –86.59 ± 5.53 mV, EE –88.56 ± 5.70 mV, $P = 0.54$; $R_N$: Ctrl 130.92 ± 38.07 MΩ, EE 114.08 ± 22.97 MΩ, $P = 0.20$). However, EE neurons generated APs at significantly lower frequencies (mean difference on average 2.4 Hz, Two-way ANOVA, $P < 0.0001$ for the factor EE, Fig. 5a). Consistent with these data, the maximum slope of the I-f curve was significantly lower in EE cells (unpaired t-test $P = 0.032$, Fig. 5b). Accordingly, we also detected a significantly higher AP current threshold in EE cells of ~70 pA (unpaired t-test $P = 0.033$, Fig. 5c, d). Other AP properties such as voltage threshold (Fig. 5c), half-width duration, and amplitude (Fig. S5a) as well as the axonal and somatic components in the phase-plane trajectories of the AP (Fig. S5b) all remained unchanged. To exclude that AIS length and therefore electrophysiological properties would change during incubation of acute slices in ACSF, we plotted the time of the start of the whole-cell recording relative to the time of slice preparation, but found no correlation in current threshold (Fig. S5c). However, consistent with the presumed role of the AIS, post-hoc correlation analysis between AIS length and threshold in biocytin-filled cells revealed that neurons with a higher current threshold also had shorter AIS ($r^2 = 0.76$, $P = 0.0005$, Fig. 5e, f). On the other hand, there was no correlation between AIS onset and current threshold ($r^2 = 0.009$, $P = 0.768$, Fig. S5d). Since our long-term (2 weeks) deprivation experiments revealed changes to the synaptic input (Fig. 3h), we recorded PSCs after exposure to 3 h EE (Fig. 5g, h). However, we did not detect any significant differences of mean amplitude or frequency of PSCs (Fig. 5h).

The above results from the relationship between c-Fos expression and AIS length suggest that increased activity of single neurons triggers homeostatic scaling in the form of an AIS length reduction. To test this link more directly, we subjected acute brain slices to 1, 3, or 6 h of elevated extracellular potassium (8 mM KCl) to induce a chronic increase in activity (Fig. 6a, c). In a second alternative approach, increased activity was induced by adding 15 μM bicuculline (Bicu) to the standard ACSF (Fig. 6a, b). Strikingly, both experiments yielded a similar result: AIS length, as analyzed in layer II/III of S1BF in acute brain slices, rapidly and significantly shortened after only 1 h of incubation in either Bicu or KCl (for Bicu, One-way ANOVA $P = 0.010$, KCl One-way ANOVA $P = 0.0044$, Fig. 6b, c). The length reductions were on average ~8 and ~10 μm for KCl and Bicu, respectively. AIS length then steadily increased again, reaching baseline levels after 6 h of incubation, thus mimicking the rapid reversal of AIS shortening that occurred in the in vivo paradigm (Fig. 4).

In summary, our data suggest that the increase in neuronal activity elicited by the exposure to increased sensory input led to a rapid reduction in AIS length, a reduced current threshold and consequently homeostatic scaling of intrinsic excitability of single neurons under behaviorally relevant and physiological conditions.

## Discussion

The present study identifies that in vivo AIS plasticity acts as a homeostatic scaling mechanism to maintain a dynamic equilibrium for individual neurons after changes in the whisker-to-barrel pathway activity. This is supported by data showing temporally distinct, bidirectional changes of AIS length and neuronal excitability after manipulating network activity in one and the same sensory system.

The rodent whisker-to-barrel system is one of the most widely studied systems to investigate the effects of sensory experience and experience-driven modulation of network state on neuronal circuit formation, not only during development but also in the adult (reviewed in[16,33,47]). The whisker-to-barrel network is active from the emergence of whiskers onward (already during the late embryonic period), and serves important functions immediately after birth in rodents ([48], reviewed in[41]). At the network level, early spontaneous whisker deflections as well as passive stimulation by the mother and littermates trigger cortical burst firing[49]. These synchronous and spatially confined spindle bursts persist during the early postnatal period[50] and begin to wear off from P3 onwards[51]. Around P12, network activity is desynchronized[52]. In keeping with this developmental time line, we found that from the time of first whisker emergence prior to birth (E20) throughout the first two weeks of postnatal development, AIS in cortical layers II/III and V gradually increase in length (Fig. 1), with layer V AIS elongating before layer II/III.

Around P12, mice start to exhibit gradually more mature rhythmic whisking behavior as indicated by higher frequencies of ~12 Hz with larger amplitudes[41]. At the same time, information processing in S1BF changes profoundly in a layer-specific manner[53]. With the sudden onset of sensory input around P12–13, and hence an elevation of cortical network activity, we find that AIS shorten, subsequently reaching an adult length (Fig. 2). This tri-phasic AIS maturation profile in the somatosensory cortex resembled, albeit less pronounced, the developmental changes in pyramidal neuron AIS in the visual cortex[28,29]. After the onset of active whisking and explorative behavior at P12–13 (ref. [54]), a reduction in AIS length was observed, followed by a gradual length decrease until adulthood. These phasic length changes coincide with several developmental programs. For example, the second postnatal week is a period of wiring of intra- and inter-layer connections in S1BF[55,56], which represents homeostatic changes. These ultimately lead to balanced network activity, especially regarding the emergences of excitation/inhibition balance in cortical networks during this time[43,57]. Also, GABAergic synapses originating from cortical chandelier cells (ChC) contact the AIS between P12 and P18 in mouse S1[58,59]. The period of AIS shortening therefore coincides with the peak of ChC synapse formation and overall increase in inhibitory conductance[60,61].

We found that long-term sensory deprivation triggers AIS elongation in layer II/III, but not V (Fig. 2), rendering layer II/III neurons more excitable (Fig. 3). The lack of sensory input for the first two weeks is most likely the direct source of these changes. In support of this idea, a study characterizing intrinsic firing properties of layer II/III neurons in rat S1BF at P12, P14, and P17 found that neuronal excitability slightly decreases with age[43]. The authors performed whisker deprivation from P9 and found no effect on the spiking properties of these neurons, suggesting that the developmental reduction in excitability is independent of sensory input and reflects e.g., membrane ion channel maturation and/or changes in morphological features. Similarly, we found that brief deprivation (P10–15), even during the peak of important critical periods for short-range synapses (layer IV to layer II/III ipsilateral) and long-range synapses (layer II/III to layer II/III contralateral;[56,62]), does not lead to significant AIS length changes (Fig. 2). However, complete deprivation from birth to P15

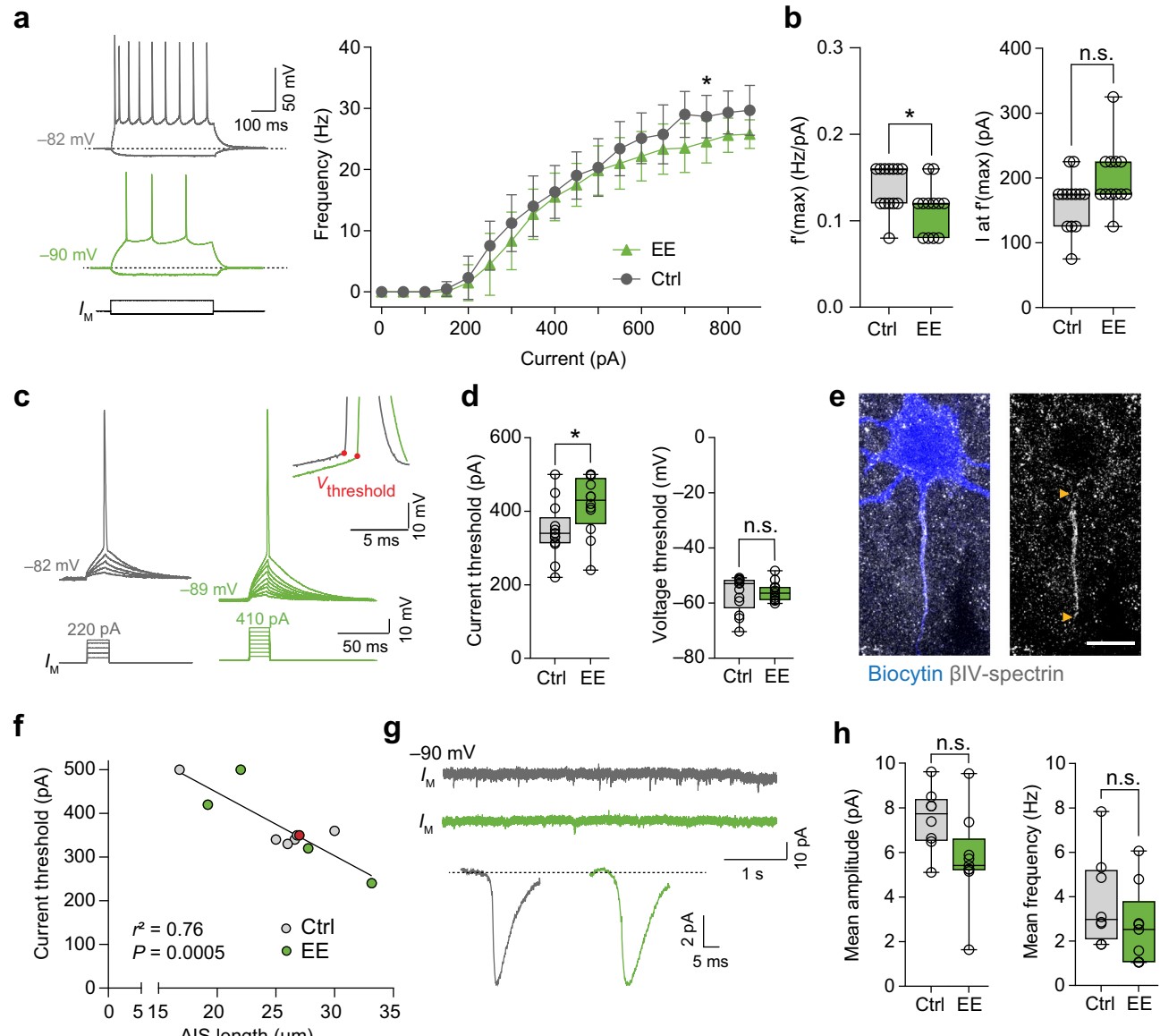

**Fig. 5 Decreased excitability of layer II/III pyramidal neurons after 3 h enriched environment (EE). a** Left: Representative traces of AP trains elicited by current injection (500 ms, −50 pA, +250pA). Note the decreased firing frequency in the EE neuron (green trace). Right: Input/frequency relationship as determined by 500 ms long injections of increasing currents. The EE group showed significantly decreased firing frequencies (Two-way ANOVA (current x EE) ***$P < 0.0001$ for current injection and EE, $P = 0.38$ for the interaction, Sidak's multiple comparison test *$P = 0.035$, $n = 13$ Ctrl cells, 12 EE cells from 10 biologically independent experiments). Data are presented as mean ± SD. **b** Analysis of maximum slope f'(max) of the I-f curve and current at the maximum slope I at f'(max) for the respective neuron. f'(max) was significantly lower in the EE group (unpaired two-sided t-test *$P = 0.032$). I at f'(max) was unchanged (unpaired two-sided t-test $P = 0.058$ $n = 13$ Ctrl cells, 12 EE cells from 10 biologically independent experiments). **c** Representative traces of single APs elicited by 20 ms current injections; 10 pA increments of injected current used to determine AP current and voltage threshold (inset). **d** Analysis of current and voltage threshold for threshold APs. EE neurons had significantly higher current threshold (unpaired two-sided t-test *$P = 0.033$). Voltage threshold was unchanged (Mann–Whitney test $P = 0.73$, $n = 13$ Ctrl cells, 12 EE cells from 10 biologically independent experiments). **e** Representative confocal image of an EE neuron filled with biocytin (blue) for AIS length analysis via colabeling with βIV-spectrin (gray). Arrowheads indicate start and end of AIS. Scale bar 10 μm. **f** Correlation analysis of relationship between AIS length and current threshold. Results of linear regression analysis indicated in figure ($n = 11$ cells from at least 6 biologically independent experiments). Red dot indicates example cell from image in **e**. **g** Representative traces of postsynaptic currents (PSCs) recorded at −90 mV. Top trace shows 6 s of recording. Bottom trace shows the averaged detected PSCs across the entire recording session of 2 min. **h** Mean amplitude and frequency of PSCs. No significant difference was observed (unpaired two-sided t-test, amplitude: $P = 0.058$, frequency: $P = 0.22$, $n = 8$ Ctrl cells, $n = 9$ EE cells from at least six biologically independent experiments). Boxplots: median with 25–75% interval, error bars show minimum to maximum values.

causes significant structural and functional plasticity of the AIS (Figs. 2 and 3). Whether intermediate deprivation windows (10 days) would be able to induce AIS plasticity remains to be examined. Nevertheless, even though deprivation was continuous throughout the early postnatal period, developmental AIS

shortening still occurred, albeit in an alleviated manner. This suggests that developmental AIS plasticity is only partially mediated by sensory input, and might additionally be driven by intrinsic, developmental programs that are genetically pre-determined. Possibly, the type of non-invasive deprivation chosen

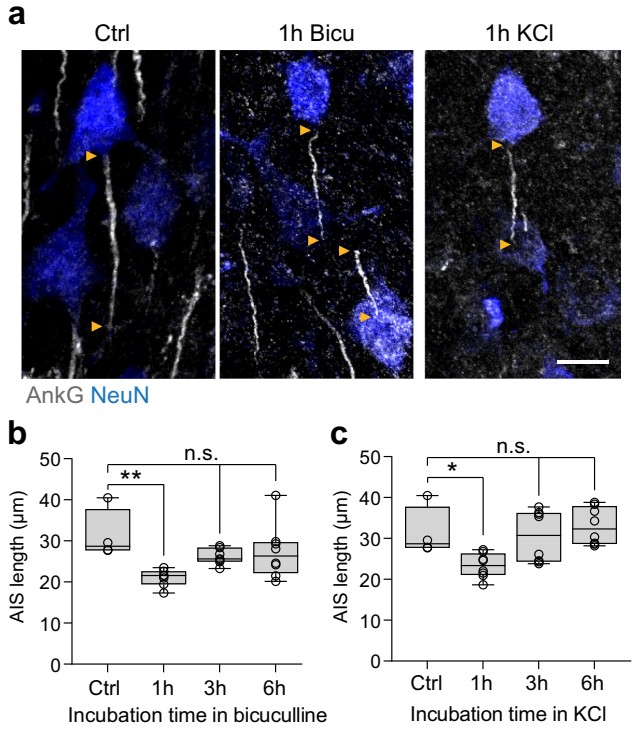

**a**

Ctrl    1h Bicu    1h KCl

AnkG NeuN

**b**

AIS length (μm)

**Incubation time in bicuculline**
Ctrl  1h  3h  6h

**c**

AIS length (μm)

**Incubation time in KCl**
Ctrl  1h  3h  6h

**Fig. 6 Rapid AIS length shortening in acute slices after 1 h of increased activity. a** Representative confocal images of layer II/III neurons in S1BF of acute slices that were subjected to 1 h of KCl (8 mM) or bicuculline (Bicu, 15 μM) treatment. Note the shorter AIS length in comparison to Ctrl conditions. Scale bar 10 μm. **b** Population analysis of AIS length in layer II/III S1BF after exposure to Bicu: 1 h leads to a shortening of AIS length (One-way ANOVA **$P = 0.010$, Dunnett's multiple comparisons test **$P = 0.0040$). AIS length returns to baseline after 3 h and 6 h of Bicu treatment. $N = 4$ slices for Ctrl, $n = 7$ independent experiments for 1, 3 h; $n = 8$ independent experiments for 6 h, slices derived from 2 mice, > 200 AIS/ condition. **c** Population analysis of AIS length in layer II/III S1BF after exposure to elevated extracellular KCl: 1 h leads to shortening of AIS length (One-way ANOVA **$P = 0.0044$, Dunnett's multiple comparisons test *$P = 0.035$). AIS length returns to baseline after 3 h and 6 h of KCl treatment. $N = 4$ biologically independent experiments for Ctrl, $n = 8$ biologically independent experiments for 1, 3, 6 h, slices derived from 2 mice, > 200 AIS/condition. Boxplots: median with 25–75% interval, error bars show minimum to maximum values.

in our experiments (touch sensors are left intact) affects the strength of overall AIS remodeling. This may allow for a remaining, albeit weak, general activation of the whisker-to-barrel pathway as opposed to methods of more complete elimination of sensory input (e.g., cauterization of the infraorbital nerve or whisker plucking). The passive touch information through pressure on the snout might already be sufficient to drive developmental, genetically-encoded programs.

We unexpectedly also observed AIS plasticity in layer II/III of adult animals when exposed to long-term whisker deprivation (Fig. 2). Also these activity-dependent changes were not observed in layer V pyramidal neuron axons. These findings are in support of the general notion that supragranular layers in S1BF retain plasticity in response to long-term deprivation into adult ages[63]. Furthermore, studies in sensory cortices of various rodent species have outlined a layer-specific ability for plastic adaptation in, for example, spine dynamics and synaptic plasticity both during development as well as at mature ages[53,64–66]. These studies showed that synaptic scaling and spine dynamics are a predominant feature of supragranular layers and to a much lesser

extent of infragranular layers, underscoring our data showing layer-specific AIS plasticity. Overall, this might reflect different roles for supra- vs. infragranular neurons during events of developmental and behavior-induced plasticity.

A key finding of the present study is the emergence of two distinct temporal scales at which AIS plasticity operates: sensory deprivation leads to AIS elongation and increased excitability if employed over periods of weeks (Figs. 2 and 3). On the other hand, AIS shortening and a decrease of excitability can be produced within an hour of sensory stimulation in vivo (Figs. 4 and 5) or via increased neuronal activity ex vivo (Fig. 6). The structural changes were consistent and correlated with bidirectional changes in AP generation, supporting the idea that the anatomical changes led to functionally relevant shifts in intrinsic excitability. The two directionally and temporally distinct forms of AIS plasticity may reflect discrete underlying molecular mechanisms. Corroborating data for a long-term plasticity mechanism have been provided by studies of the avian and mammalian auditory system, where at least one week of deprivation was required in order to elicit AIS elongation and changes in neuronal excitability[11,30]. Interestingly, in the absence of a change in the gap between soma and the proximal AIS (Figs. S2c and S4c), our data suggests that elongation seems to occur at the distal AIS.

In contrast, rapid AIS plasticity as reported after significant elevation of neuronal activity in vitro occurs on a much faster time scale[12,19,23]. Here, we found that rapid AIS plasticity takes place after a brief increase in activity during a behaviorally relevant context in vivo, with AIS length reduction already 1 h after exposure to EE (Figs. 4 and 5). Similar results from acute slices exposed to KCl or bicuculline further support our hypothesis that AIS disassembly is generated by elevated neuronal activity (Fig. 6). Together with the deprivation induced AIS length changes, the rapid AIS length reduction provides evidence of an ongoing homeostatic mechanism at the AIS to normalize activity in reaction to changes in network activity. Unexpectedly, shortened AIS returned to their baseline length (i.e., "lengthened") within the 3 h after returning to their home cage (Fig. 4e).

What are the possible molecular mechanisms that drive bidirectional slow and rapid structural AIS plasticity? And how could the rapid re-elongation after initial length reduction be explained if AIS length increase after sensory deprivation requires many days? Our data indicates that sensory deprivation triggers AIS elongation at the distal end. Such an elongation would require de novo protein synthesis outside of the axon, and transport of macromolecular complexes via the microtubule network towards the distal AIS site, with subsequent assembly and integration of AnkG and other AIS proteins into the axolemma[67]. This is presumably a time-consuming process, particularly considering the vast downstream network of cytoskeletal proteins and molecular binding partners localized in the AIS[68]. Indeed, de novo assembly of an immature AIS in vivo takes several days[69] and is consistent with our observation that many days to weeks of deprivation are necessary to elongate the AIS. In contrast, rapid AIS shortening after just 1 h of altered network activity is possibly associated with AnkG protein internalization and altered transport, rather than actual degradation, thereby allowing the re-assembly of the AIS scaffold in a notably faster timeframe. Although the specific molecular pathways of AIS reassembly remain to be examined experimentally, evidence for internalization rather than degradation is provided by the finding that in vitro AIS shortening elicited by high extracellular KCl is calpain-independent and therefore not promoted by proteolysis[23]. Rapid AIS elongation within the timeframe of hours may thus be possible because the AIS is re-assembled with AIS proteins that were not degraded during the initial reduction, but

rather internalized. Future studies with live imaging would be required to examine these pathways directly.

Our analysis of AIS length in c-Fos positive versus c-Fos negative neurons (Fig. 4g) demonstrates that AIS plasticity occurs selectively in those neurons that receive the strongest sensory activation, thus establishing a link between network activity and AIS plasticity. These results support our hypothesis that rapid AIS plasticity operates as a homeostatic scaling mechanism for activity regulation at a cell-autonomous level. Mechanistically, previous studies have established that the coupling of activity to AIS remodeling is most likely mediated by activity-triggered $Ca^{2+}$ signaling pathways. One of the potential downstream actors is the $Ca^{2+}$-dependent phosphatase calcineurin. Activity-dependent AIS plasticity is reduced or even completely abolished after blockage of either L-type $Ca^{2+}$ channels or calcineurin directly[12,23]. Future studies are required to shed light on the interplay of the various regulatory proteins at the molecular level. Interestingly, both in the long-term deprivation experiments as well as after EE, only AIS length but not AIS onset location changed and was a strong predictor of AP current threshold (Figs. 3, 5, S3 and S5). These findings were consistent with the observed changes in somatically recorded AP properties. The first component in the AP, which has been shown to be determined, in part, by AIS onset[37,70], was not affected either. AIS length changes were selectively associated with the current to reach threshold, consistent with theoretical predictions, modeling, and experimental studies predicting that AIS length scales the current threshold (reviewed in[17]). Based on these studies, an increase in AIS length will increase the total number of sodium channels increasing the magnitude of transmembrane and axial current flow and is likely associated with a more distal AP initiation site. Both factors will contribute to a reduction in the rheobase current for AP generation. The input current-frequency plots in the deprivation group confirmed that excitability was increased around current threshold levels, whereas maximum firing frequencies remained unaffected. Our data show that after deprivation, synaptic input in layer II/III became strengthened and more frequent as indicated by the spontaneous postsynaptic currents. These results differ from previous studies showing that whisker deprivation weakens excitatory layer IV to layer II/III input, while layer IV neuronal responses remain unchanged in rat S1[63,71]. We cannot exclude cell-intrinsic changes in layer IV neurons in our model following the whisker deprivation. Further studies will need to investigate whether AIS plasticity also occurs in layer IV as well as at preceding thalamic relay stations.

In summary, our study provides evidence for a direct role of AIS plasticity in homeostatic scaling mechanisms in S1BF during development and behaviorally-evoked network changes within physiological contexts in vivo. AIS plasticity appears to be a direct response to changes in neuronal activity and occurs in an unexpectedly short timescale. Furthermore, evidence for a homeostatic role is provided by data indicating that AIS plasticity is bidirectional, reversible, and thus a continuously ongoing adaptation. Further experiments are required to understand the molecular mechanisms to scale and optimize the AIS and being able to abolish or accelerate the AIS plasticity responses by cell-type-specific molecular means. Obtaining such insights and addressing these mechanistic questions would require live-imaging of single AIS ex vivo or in vivo. Once adequate live AIS reporters are available for in vivo application, such studies will provide exciting new avenues to further our understanding of AIS plasticity in physiological contexts.

## Methods

**Animals**. All animal procedures were carried out in accordance with the recommendations of the Animal Research Council of the Medical Faculty Mannheim,

Heidelberg University and were approved by the State of Baden-Württemberg (35-9185.81/G-242/12, 35-9185.81/G-290/16 and 35-9185.81/G-119/20) and compliant with EU guidelines. All experiments were conducted using wildtype mice (C57BL/6JRj obtained from Janvier Labs, France). Animals were randomly picked for the different experimental and control groups, and all groups consisted of animals from both genders. All animals were maintained with food and water ad libitum on a regular 12 h light/dark cycle in ambient light conditions (300 lux during light, 25 lux in the resting section of all cages). Humidity was kept at 50% (±10%) and room temperature was 22 °C (±2 °C). Electrophysiological experiments were conducted between 3 and 8 h after onset of the light phase. Tissue from PV-tdTomato animals for control immunofluorescence was provided by Jochen Staiger, University of Göttingen, Germany.

**Developmental and deprivation study**. For the analysis of barrel cortex development, a total of 5–6 brains were analyzed in each of the following age groups: E20, P1, P3, P7, P10, P13, P15, P21, P28, P35, P45, P180. The maturation of AIS length is a robust indicator of developmental progression[28,29,72] and was therefore chosen as the key AIS parameter in this study. At least 100 AIS per animal were examined in S1BF in layers II/III and V, respectively. For some of the age points, an additional three animals were sacrificed for Western blot analysis of AIS scaffolding protein expression.

For sensory deprivation experiments, animals were subjected to daily bilateral whisker trimming using a curved eye scissor from P0, P10 or >P100 to different end points (see Table S1 and Fig. 2a). Whisker regrowth was monitored daily under a binocular microscope and constantly kept below 1 mm of length. In some experimental groups, whiskers were allowed to regrow (see Table S1 and Fig. 2a). Animals older than P15 were briefly anesthetized with isoflurane prior to handling. Adult mice (>P100) were anesthetized with 40 mg Ketamine/ 5 mg Xylazine i.p. to minimize stress during trimming. To exclude any effects of handling or anesthesia on AIS length, control experiments were performed (see Table S3 and Fig. S2c). Electrophysiological measurements were performed in 7 animals per group, yielding about 15–20 cells per group and analysis.

**Enriched Environment**. P28 mice were whisker-trimmed unilaterally on the evening prior to the experiment so that the corresponding hemisphere was serving as an internal control (Fig. 4a). 12 h later, mice were placed in a large cage containing different types of bedding, wall textures, and various novel objects for either 0, 1, 3 or 6 h, and the cage was placed in the dark to increase explorative behavior (Figs. 4 and 5). To control for any effects of the unilateral whisker trimming, one group was trimmed and perfused the next morning without being placed in an enriched environment (0 h group, Fig. 4d). No significant changes between the two hemispheres were observed in the 0 h group. This allowed us to use the hemisphere corresponding to the trimmed whisker pad as a control for the hemisphere experiencing the increased sensory input within the same animal. Immediately after the indicated times, animals were either subjected to cardiac perfusion for immunofluorescence or sacrificed for electrophysiological recordings (see section below and Table S4). One experimental group was placed back in a normal home-cage for an additional 3 h before being sacrificed (Fig. 4e). Another experimental group was placed into a new EE (with novel objects and bedding) for an additional 1 h after 6 h of the first EE. We included 5–6 animals per group for immunofluorescence and 10 per group for electrophysiology ($n = 13$ cells Ctrl, 11–12 cells EE).

**Immunofluorescence**. For the developmental and deprivation studies, P0–P7 animals were decapitated and brains were dissected in ice-cold 0.1 M phosphate buffer (PBS), fixed for 5 min by immersion in 4% paraformaldehyde (PFA, in 0.1 M PBS, pH 7.4) at 4 °C and cryoprotected in 10% sucrose (overnight), followed by 30% sucrose (overnight) at 4 °C. Animals P10 and older were exsanguinated with 0.9% NaCl under deep anesthesia with Ketamine (120 mg/kg BW)/Xylazine (16 mg/kg BW) and perfusion-fixed with ice-cold 4% PFA for five minutes. Brains were then removed from the skull and were cryoprotected in 10% sucrose (overnight) followed by 30% sucrose (overnight) at 4 °C. Tissue was trimmed to a block including S1BF and embedded in Tissue Tek (Sakura Finetek). Multichannel immunofluorescence staining was performed on 20 μm sections collected directly on slides[28]. Slices were incubated in blocking buffer (1% BSA, 0.2% fish skin gelatine, 0.1% Triton in 0.1 M PBS) for at least 60 min and subsequently incubated in primary antibodies overnight at 4 °C. After washing, slices were incubated for at least 90 min in secondary antibodies in the dark at room temperature. In some cases, the nuclear stain TO-PRO-3 Iodide (1:1000, Thermo Fisher) was added to the last washing step after secondary antibody incubation. For preservation of immunofluorescence, slices were mounted in a mounting medium with anti-fading effect (Roti-Mount FluorCare, Carl Roth). After omission of the primary antibodies and application of only secondary antibodies, no specific immunolabeling was observed. Antibodies were further tested for validity as outlined in Table S5.

Acute slices from electrophysiological recordings were fixed in 4% PFA for 20 min. Staining was carried out as described above, however with prolonged incubation periods (blocking for 2 h, primary antibody 2× overnight for total of 48 h, secondary antibody 1× overnight) to allow for sufficient penetration of the tissue by the antibody. We frequently observed that in biocytin-filled neurons, AIS

stainings appeared weaker. Consequently, only AIS with clearly detectable start and end points were chosen for the length correlation analysis (Figs. 3f and 5d).

**Image acquisition and analysis**. Confocal analysis was carried out on a C2 Nikon confocal microscope (Nikon Instruments, laser lines: 642, 543, and 488 nm) with a 60× (oil immersion, NA 1.4) and 100× objective (oil immersion, NA 1.45), respectively, and a SP5 confocal microscope (Leica, Mannheim; laser lines: 633, 561, 514, and 488 nm) with a 63× objective (oil immersion, NA 1.4). To increase the number of in-focus immunoreactive structures, stacks of images were merged for analysis (maximum intensity projection). For visualization of representative images in figures, brightness and contrast were optimized (ImageJ). Thickness of single optical sections was 0.5 μm in stacks of 10–20 μm total depth for cryosections and up to 40 μm in acute slices. Confocal x-y-resolution was set to and kept at 0.21 μm (60×) or 0.12 μm (100×) per pixel. Images for qualitative analysis were evaluated and enhanced for brightness and contrast in Photoshop (Adobe Systems) and FIJI (ImageJ), respectively. AIS length was measured using a self-written macro[28,38] as well as the morphometrical software AISuite[73]. This tool extends the well-established and widely used method of defining AIS start and end points as points where a predefined fluorescence threshold (relative to the maximum fluorescence intensity along a line drawn over an individual AIS) is surpassed[10]. The threshold was adjusted depending on the individual staining quality and ranged from 10 to 30% of maximum fluorescence intensity. Both analysis tools were tested for inter-method reliability, revealing robust consistency of results.

**Western blot**. For mice from E20.5 - P3, processed samples included the entire cortex. For older animals, all brains were cut into 1 mm slices using a tissue matrix slicer (Zivic instruments) and single sections were visualized using a binoscope to carefully dissect only S1BF for sample preparation. Additionally, animals older than P10 were perfused transcardially with ice-cold 0.9% NaCl as outlined above before removing the brain. Samples were diluted in a homogenization buffer (20 mM Tris, 0.5 M NaCl, 8 mM CHAPS, 6.4 mM EDTA, pH 7.5) containing phosphatase and protease inhibitors (Sigma-Aldrich). Samples were then homogenized via ultrasonication, lyzed for 60 min and centrifuged at $16.2 \times g$ for 45 min at $-4$ °C. Protein quantification via a Bradford assay was performed and 20 μg samples containing Laemmli-buffer (2% SDS, 60 mM tris-Cl, 10% glycerol, 5% ß-mercaptoethanol, 0.01% bromphenol blue) were heated for 10 min at 70 °C. Gradient gels (3–8% Tris-acetate protein gels, Thermo Fisher Scientific) were loaded with the lysates and run for 55 min at 150 V in Tris-tricine buffer (50 mM Tris, 50 mM tricine, 0.1% SDS). The blotting process was carried out in three steps[74]. This ensured transfer of both large and small proteins on the same membrane so ankG isoforms (ranging from 190 to 480 kDa) as well as the loading control actin (50 kDa) could be transferred and visualized for later analysis. Blotting was performed at 550 pA in a tris-glycine buffer (25 mM tris-Base, 192 mM glycine) under constant cooling. Solutions contained 20% methanol for 30 min, 15% methanol and 0.05% SDS for 30 min, and only 0.1% SDS for another 90 min. Between each step, membrane strips already containing smaller proteins were removed. Membranes were blocked for 60 min in PBST (protein free). Primary antibodies were incubated overnight at 4 °C (summarized in Table S5) and secondary antibodies were incubated for 90 min at room temperature. Protein signal was revealed with an ECL Kit (Western Bright ECl HRP substrate, Advansta) and imaged (Fusion solo, Vilber Lourmat). Analysis was carried out with Image J software. Samples were normalized against the internal loading control (actin) as well as, when comparing several gels, against a standardized sample with a consistent protein concentration run on each gel.

**Electrophysiology**. In accordance with the guidelines of 3Rs, and to enable the use of several animals per litter, animal age for Fig. 3 (Group 1) ranged from P13 to P16 (both for Dep and Ctrl) and animal age for Fig. 5 ranged from P28 to P31 (EE). Mice were briefly anesthetized with isoflurane (3%) and decapitated. The brain was quickly removed and placed in ice-cold sucrose-based cutting solution (206 mM sucrose, 2.5 mM KCl, 1.25 mM NaH$_2$P0$_4$, 25 mM NaHCO$_3$, 25 mM Glucose, 3 mM MgCl, 1 mM CaCl$_2$, pH 7.4), which was saturated with carbogen (95% O$_2$, 5% CO$_2$). 300 μm thick coronal sections containing S1BF were cut with a vibratome (VT 1200S, Leica Biosystems). Acute slices were transferred to artificial cerebrospinal fluid (ACSF; 125 mM NaCl, 2.5 mM KCl, 1.25 mM NaH$_2$PO$_4$, 25 mM NaHCO$_3$, 1 mM MgCl$_2$, 2 mM CaCl$_2$, 25 mM glucose, pH 7.4, oxygen-saturated with 95% O$_2$, 5% CO$_2$) and allowed to rest at room temperature for at least 15 min (EE conditions) and 30 min (deprivation conditions) before recordings began. All recordings were carried out at room temperature. For EE experiments, time after slice preparation was kept to a minimum and was monitored to exclude any rapid reversal of AIS shortening during incubation of acute slices (Fig. S5c). Slices were imaged with an upright Nikon Eclipse FN1 equipped with a DIC contrast filter. Layer II/III pyramidal neurons were visually identified and neuron type was confirmed post-hoc by the firing pattern and immunofluorescence against βIV-spectrin and the biocytin fill. Pipettes were pulled from borosilicate glass (outer diameter 1.5 mm, inner diameter 0.8 mm, Science Products) to a tip resistance of 3.5–5.5 MΩ and filled with intracellular solution (140 mM K-gluconate, 3 mM KCl, 4 mM NaCl, 10 mM HEPES, 0.2 mM EGTA, 2 mM Mg-ATP, 0.1 mM Na$_2$-GTP), containing 3 mg/ml biocytin. Patch-clamp recordings were made with a HEKA

EPC10 USB amplifier controlled by Patchmaster Software (HEKA Electronics). Signals were filtered with a 10 kHz (Filter 1) and 2.9 kHz (Filter 2) Bessel filter, digitized, and sampled at 50 kHz. The liquid junction potential was calculated to be $-12$ mV and corrected for post-hoc. Fast and slow capacitances were compensated for in cell-attached and whole-cell configuration, respectively. Series resistance ($R_s$) was constantly monitored with a $-10$ mV step in voltage clamp. Cells with $R_s$ exceeding 30 MΩ during recordings were excluded from analysis. $R_s$ was not changed between the groups (P15 Ctrl 14.88 ± 2.62 MΩ, P15 Dep 14.73 ± 5.04 MΩ, $P = 0.92$; Ctrl 21.81 ± 6.87 MΩ, EE 21.73 ± 5.86 MΩ, $P = 0.98$). Input resistance ($R_N$) was calculated from the slope of the current/voltage relationship curve from a current clamp step protocol. Resting membrane potential (RMP) was measured directly upon entering whole-cell configuration in current clamp at $I = 0$. AP properties were measured with a step protocol of 20 ms pulses increasing in 10 pA increments, starting from a holding current of $I = 0$. For firing pattern analysis (including the I-f curves), 500-ms long pulses incrementing in 50 pA steps were used to trigger AP trains. Maximum slope of the curve as well as the current at maximum slope were calculated for each group. Spontaneous postsynaptic currents (PSCs) were recorded at $-90$ mV for 2 min. Since the recording voltage of $-90$ mV is more hyperpolarized than the chloride reversal potential of approximately $-70$ mV in our conditions, recorded currents were presumed to be a mix of excitatory and inhibitory synaptic currents.

Analyses were carried out offline with either FitMaster Software (HEKA Electronics) or OriginPro 8 (Origin lab Corporation). Current threshold was defined as the current at the first 20 ms pulse that reliably elicited an AP. Voltage threshold was determined as the point where the first time derivative exceeded 50 V s$^{-1}$. AP amplitude was measured from voltage threshold to the AP peak voltage. AP half-width was defined as the width at the middle voltage of the rising phase between AP threshold and peak. For phase plot analysis, the first temporal derivative (V s$^{-1}$) was plotted against the voltage (V), and the value at the first (AIS) and second (somatic) peak of the AP were extracted for analysis. PSCs were detected with the automatic event detection function of AxoGraph X (AxoGraph Scientific), and mean amplitudes and frequency were calculated for each neuron.

**Statistics and reproducibility**. Mean values and standard deviation (SD) of AIS length were calculated, plotted and analyzed in GraphPad Prism 8 software (GraphPad Software, Inc.). All data sets were individually tested for normal distribution (Shapiro–Wilk test). Unpaired two-sided t-test and Mann–Whitney test were carried out for parametric and non-parametric comparison of only two groups, respectively. Two-way ANOVA followed by appropriate post-hoc correction was applied when comparing two or more groups over several time points (details are given in the respective figure legends) for normally distributed data. All statistics were carried out at 95% confidence intervals, therefore a significant threshold of $p < 0.05$ was used in all analyses. In all graphs, box plots indicate the median (middle line) across all biologically independent experiments with min and max value (whiskers) and 25 and 75 percentiles (bottom and top border of box). P values and number of samples are stated in each figure legend.

All representative images or micrographs were obtained from biologically independent experiments repeated at least three times. A minimum of six confocal images (three in layer II/III, three in layer V) were produced in each mouse and hemisphere, spanning the entire rostro-caudal axis of S1BF. From these 6 images, at least 100 AIS per animal and condition were included in statistical analysis. Similarly, Western blot analysis and electrophysiology were performed in biologically independent experiments, the exact number of which are stated in each figure legend. Thus, replication under consideration of animal welfare and the 3R principles was achieved when considering individual mice (with at least 100 data points analyzed per mouse) as individual samples, with 5–6 mice per experimental and control groups.

**Reporting summary**. Further information on research design is available in the Nature Research Reporting Summary linked to this article.

## Data availability

Additional data that support the findings of this study are available from the corresponding author upon reasonable request. Source data are provided with this paper.

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

## Acknowledgements

The authors wish to thank Silke Vorwald for outstanding laboratory support and Johannes Roos for programming the AISuite software. Furthermore, we are indebted to Rudolph Schubert, Jana Maurer, and Martin Kaiser for providing expertise, support and technical equipment for establishing the electrophysiological experiments. We also thank Paul Jenkins and Vann Bennett for their intellectual and technical support regarding Ankyrin-G immunoblotting. We would like to thank Petra Wahle and Amélie Fréal for valuable discussion and input during the writing process of the manuscript. This work was supported by the Core Facility Life Cell Imaging Mannheim (LIMA) at the MCTN (DFG-INST 91027/10-1 FUGG). The study was funded by the German Research Foundation (DFG, SFB 1134, TP A/03) to C.S. and M.E., EN 1240/2-1 to M.E., and the Netherlands Research Council (NWO *Vici* grant 865.17.003) provided to M.H.P.K.

## Author contributions

Conceptualization, N.J. and M.E.; Methodology, N.J., R.W., J.S., and M.E.; Investigation, N.J., DD., N.L., C.T., and M.E.; Writing – Original Draft, N.J., M.H.P.K., and M.E.; Writing – Editing and Feedback, N.J., J.S., M.H.P.K., and M.E.; Visualization – N.J., M.H.P.K., and M.E.; Resources, N.J., D.D., N.L., C.T., J.S., and M.E.; Supervision, M.E.; Funding Acquisition, C.S., M.H.P.K., and M.E. All authors commented on the manuscript and approved the final version of the paper.

## Funding

## Competing interests

The authors declare no competing interests.
