## [Peer Review File · Nature Communications]

Reviewers' comments:

Reviewer #1 (Remarks to the Author):

This paper revealed homeostatic structural AIS plasticity and associated changes of neuronal excitability at layer 2/3 of mice somatosensory barrel cortex after modifications of sensory inputs in behaviorally relevant context. It further showed that this plasticity was specific to the layer, since neither AIS length nor neuronal excitability changed at layer 5 with the same sensory modifications.

The novelty of this paper is that the exposure of animals to enriched environment caused AIS length decrease, which presumably occurred via an increase of sensory inputs. This was nicely supported by the shorter AIS in c-Fos positive than c-Fos negative cells, strengthening the link between the AIS plasticity and the animal behavior.

This work was done by leading researchers in the field. Experiments were technically sound and good in quality. Discussion was concise and logically clear. The results should advance the current understanding on regulation of excitability in the cortex and be valuable for broad readers in the field of neuroscience. However, there were some concerns on the analyses and the interpretation of data, which might critically affect the conclusion. These and other comments were listed.

Major comments

1. Excitatory and inhibitory cells were reported to differ in the phenotype of AIS plasticity (Chand et al., 2015). However, cell types were not determined in the immunohistochemistry, which might underestimate the measurements. It is also important to discuss the cell type specificity of AIS plasticity in the context of homeostatic plasticity.

2. Since AIS length decrease is known to accompany sodium current increase via dephosphorylation of sodium channels (Evans et al., 2015), whether and how this contributed to neuronal excitability during each AIS plasticity should be determined.

3. Correlation between AIS length and threshold current in the same neurons was very good (Figures 3 and 5), but these parameters did not seem to differ between control and manipulated groups, raising questions for the robustness of AIS plasticity and its impact on neuronal excitability.

4. AIS length and c-Fos immunoreactivity returned to baseline at six hours (Figure 4), but it was uncertain whether it occurred due to a decrease of sensory inputs or to a decrease of neuronal excitability by the AIS plasticity. This would be an important point in understanding the homeostatic roles of AIS plasticity.

Minor comments

1. Presentation of spike traces in Figures 3 and 5 may be reversed for control and manipulated groups, because the order is opposite in other panels.

2. In Figures S2 and S4, the first component of action potential phase plot did not differ between control and manipulated groups. Does this mean that factors other than sodium current, such as potassium current, were involved in the regulation of neuronal excitability?

3. Related to the above issue, it is better to explain how the increase of AIS length could reduce the threshold current.

4. AIS length increased with long-term sensory deprivation (> fifteen days), but not with short-term one (five days). On the other hand, AIS length decreased within one hour. Why was the time course so different between the manipulations? It may also be interesting to discuss the functional implication of this variation.

5. Why was AIS apparently faint and wavy after treating slices with KCl or bicuculline? This may imply that the manipulations affected the organization of the entire AIS and/or the axon in addition to the AIS length.

6. It was not explained why AIS length returned to baseline at three hours after the treatment of KCl or bicuculline.

Reviewer #2 (Remarks to the Author):

In their study, Jamann et al. reveal AIS plasticity in layer 2/3 of the mouse barrel cortex using two different in vivo manipulations: whisker deprivation as well as sensory enrichment. By doing so they demonstrate that the AIS can physically increase as well as decrease its length in a homeostatic manner under these conditions.

The authors first show the developmental time course of AIS length changes in layers 2/3 as well as 5. Overall, AISs seem to grow until around P10-P15, then shorten again a bit before stabilising. Interestingly, the time period of AIS shortening coincides with the onset of active exploration and whisking, raising the question whether this is an activity-dependent and not merely a genetically predetermined process. To address this, the authors next turned to whisker trimming in order to reduce sensory input. This resulted in longer AIS lengths, even in the adult cortex. Electrophysiological recordings confirmed that the physical AIS changes correlated with a functional decrease in current threshold. This strongly suggests that the AIS is undergoing homeostatic plasticity in response to reduced synaptic inputs caused by the manipulation, independent of developmental stage. A common criterion for establishing a plasticity as homeostatic is bidirectionality. Here, Jamann et al. are able to demonstrate that AISs in layer 2/3 of the barrel cortex are also able to undergo shortening by using enriched environment to increase sensory input. As before, this form of AIS plasticity was accompanied by a change in current threshold. Interestingly, unlike sensory deprivation, this manipulation resulted in rapid structural changes, with a significant change already observed after 1 hour instead of after 2 weeks, and an equally fast recovery of the AIS length back to baseline after only 6 hours. This rapid time course of AIS length changes was replicated in a final experiment in vitro.

Overall, this is an interesting and compelling study that probes AIS plasticity using in vivo manipulations, demonstrating that the AIS can increase as well as decrease its length in a homeostatic context. In particular, the fast AIS shortening after sensory deprivation is of significant scientific impact, as it suggests, for the first time, that AIS plasticity is an ongoing process utilised during normal behaviour in vivo.

There are a few places where further clarifications will be beneficial, as outlined below. To convincingly support their findings, the authors should address the following, specific comments:

1) The short-term plasticity seen after sensory enrichment is particularly exciting, as it suggests that AIS plasticity is an ongoing, relatively dynamic process in vivo. In light of this it would be interesting to understand what happens at the 6 hour time-point, when AIS length returns to baseline. Does this happen because the enriched environment is simply no longer novel and thus the whisking activity has returned to normal? While returning animals to their home cage after 3 hours addresses this point to some extent, it would be helpful to know what happens if the animals were placed into a novel enriched environment after 3 hours.

The in vitro data doesn't sufficiently address this, especially without any measure of activity levels within the slices. These could have returned to normal after 3-6 hours, due to e.g. other homeostatic processes, slice deterioration, etc.

In addition, a thorough discussion of this point appears to be lacking. Do the authors suggest that the return to baseline is an intrinsic property of rapid AIS shortening, rather than activity-dependent?

2) For this set of experiments, Jamann et al. have chosen to use as control the opposite hemisphere, after whisker trimming, presumably to reduce inter-animal variability. Particularly compelling is the finding that *cfos*⁺ neurons contralateral to the stimulated side show AIS plasticity, whereas *cfos*⁻ neurons don't. The authors should consider adding data from ipsilateral (control) *cfos*⁺ and *cfos*⁻ neurons.

3) The authors interpret their data as evidence that AIS lengthening only occurs over prolonged periods of time, whereas AIS shortening can be achieved within an hour. They suggest that this might be due to complex molecular interactions and energy-consuming protein assembly and transport. However, after AIS shortening they observe an equally rapid return of AIS length to baseline values, which suggests that the AIS can lengthen quickly, at least from a shortened state. It would be helpful if the authors could add this point to the discussion, together with possible reasons for this disconnect in plasticity phenotypes.

4) Finally, while the electrophysiological changes seen in this study are small, the correlation of AIS length with current threshold is very powerful. Since AIS plasticity can manifest as both a change in length as well as a change in position, it would be interesting to also see the correlation of this with current threshold.

Minor points:

Figure 1: It would be helpful if the authors could add an example AIS image for the E20 time point.

On page 5, the authors state: "Bilateral whisker trimming resulted in a significant lengthening of AIS at all time points in Group 1 in layers II/III (...), but not in layer V (Fig S1A)."

Figure S1A doesn't actually show all time points from Group 1, hence this statement is misleading.

Throughout the main text, synaptic currents are merely called PSCs, whereas it is clear from their methods that they are recording EPSCs rather than a mix of inhibitory and excitatory PSCs. This should be changed to remove ambiguity.

Figure 6: The figure legend states that 3-4 slices were used per condition, however there are clearly more than 4 points on the plots in both 6B and 6C.

On page 16, the authors state: "After the onset of active whisking [...], a reduction in AIS length was observed, followed by a gradual length increase throughout adulthood." I cannot see any evidence for the quoted length increase. Please clarify.

Page 25, methods, the authors state: "Staining was carried out as described above, however with prolonged incubation periods to allow for sufficient penetration of the tissue by the antibody". Please specify the length of this 'prolonged incubation period'.

Reviewer #3 (Remarks to the Author):

The paper describes axon initial segment (AIS) plasticity in the somatosensory system using the relevant behavioural tasks. They show bidirectional AIS plasticity in this system and also test electrophysiological properties of neurons endured AIS structural plasticity, confirming physiological adaptation in response to the structural plasticity. They show this plasticity is restricted to the layer 2/3 neurons and not happening in layer 5 pyramidal neurons. Also, this study characterises

developmental AIS plasticity in the barrel cortex and altogether add to the previously reported AIS plasticity in the visual and auditory systems. Experimenting of AIS in C-Fos activated neurons is a major advantage of this study confirming the plasticity as a result of changes in activity at single neuron level, basically linking in vitro data to the in vivo data in a practical way.

The findings are novel for the somatosensory system and in line with the previous reports in visual and auditory system with minor system-specific differences. The methodical and analytical approach are very clear and precise and covers a range of technical expertise, all well-executed.

A few minor points;

It is not clear why β IV-spectrin has been used for AIS length measurement and AnK G for immunoblot analysis? Why not use one of them for both for consistency? Any technical reason?

I am not convinced of the conclusion that 15 days of deprivation is necessary for AIS plasticity to happen, the only experiment for this conclusion is data obtained from group 2. What if the 10 days deprivation would be between 10-20 days? My argument is not this additional group to be done, as the study does a thorough check of the different windows already, but this part of conclusion should be toned down.

In supplementary tables, there are typos in regard to RN and RS unit of measurements, ($M\Omega$)

Discussion in general is good and has addressed most of the findings but has missed on some of the literature on AIS plasticity, in particular the only reference available on the effects of enriched environment on AIS length (Nozari et al., Dev Psychobiol, 2017). It would be very interesting to see if the reported AIS plasticity in the primary somatosensory cortex can be communicated to the secondary sensory areas or association areas and at least the evidence can be discussed.

Response to Reviewers' comments

Reviewer #1:

This paper revealed homeostatic structural AIS plasticity and associated changes of neuronal excitability at layer 2/3 of mice somatosensory barrel cortex after modifications of sensory inputs in behaviorally relevant context. It further showed that this plasticity was specific to the layer, since neither AIS length nor neuronal excitability changed at layer 5 with the same sensory modifications.

The novelty of this paper is that the exposure of animals to enriched environment caused AIS length decrease, which presumably occurred via an increase of sensory inputs. This was nicely supported by the shorter AIS in c-Fos positive than c-Fos negative cells, strengthening the link between the AIS plasticity and the animal behavior.

This work was done by leading researchers in the field. Experiments were technically sound and good in quality. Discussion was concise and logically clear. The results should advance the current understanding on regulation of excitability in the cortex and be valuable for broad readers in the field of neuroscience. However, there were some concerns on the analyses and the interpretation of data, which might critically affect the conclusion. These and other comments were listed.

Major comments

1. Excitatory and inhibitory cells were reported to differ in the phenotype of AIS plasticity (Chand et al., 2015). However, cell types were not determined in the immunohistochemistry, which might underestimate the measurements. It is also important to discuss the cell type specificity of AIS plasticity in the context of homeostatic plasticity.

We thank the reviewer for raising this issue. We always took care to ensure that the predominant cell type we analyzed was a pyramidal cell. First, only AIS perpendicular to the pial surface of the slice were analyzed, and second, only AIS of equal diameter were considered for each developmental and adult age. From our own previous work, we have a good understanding of the differences in orientation of AIS when comparing pyramidal cells and interneuron populations (Höfflin et al., 2017), which harbor AIS of mostly non-perpendicular orientation. Furthermore, also the diameter is notably smaller than that of neighboring pyramidal AIS. Finally, the AIS signals in interneurons are notably tubular, while AIS signals in pyramidal neurons show a diameter decrease from axon hillock into the distal axon. We have added Fig. S1B in the supplements highlighting these differences using representative images from AIS immunostaining in an adult transgenic Parvalbumin-tdTomato mouse.

2. Since AIS length decrease is known to accompany sodium current increase via dephosphorylation of sodium channels (Evans et al., 2015), whether and how this contributed to neuronal excitability during each AIS plasticity should be determined.

The reviewer correctly points out that under some experimental conditions, AIS length changes together with sodium channel phosphorylation is observed (Evans et al., 2015). However, in that specific study, these diverging actions of the AIS became apparent since the AIS length change itself was not accompanied with a reduction in neuronal excitability, and dephosphorylation masked the AIS length modifications. In the present study, we instead find a significant negative correlation between AIS length and current threshold under physiological conditions (Figs. 3D and 5D). While this does not completely rule out the possibility that sodium channel phosphorylation occurs, if this happens, it will play a minimal role.

3. Correlation between AIS length and threshold current in the same neurons was very good (Figures 3 and 5), but these parameters did not seem to differ between control and manipulated groups, raising questions for the robustness of AIS plasticity and its impact on neuronal excitability.

We agree with the reviewer that both groups in the electrophysiological experiments are heterogeneous in measured length and current threshold (Fig. 3D). Because of the relatively low n -values in electrophysiological experiments, our recorded neurons may not necessarily represent the full parameter range in the population (Fig. 2E). Despite this methodological limitation, however, it is important to note that the key differences between the two groups were preserved: i) the current threshold was significantly lower (Fig. 3C), and ii) when comparing mean AIS length in the deprived neurons with control neurons, their AIS were significantly longer (Mann-Whitney test $P = 0.032$). Additionally, our correlation analysis shows that there is a significant effect of AIS length on current threshold ($r^2 = 0.36$, $P = 0.029$, Fig. 3D). The interdependence of these parameters strongly supports our hypothesis that AIS plasticity affects neuronal excitability.

4. AIS length and c-Fos immunoreactivity returned to baseline at six hours (Figure 4), but it was uncertain whether it occurred due to a decrease of sensory inputs or to a decrease of neuronal excitability by the AIS plasticity. This would be an important point in understanding the homeostatic roles of AIS plasticity.

We agree with the reviewer this is an important experiment. In the revised manuscript, we have now included data from an experiment in which mice were placed for 6 hours in an enriched environment followed by 1 hour in a new enriched environment. The results showed that 1 h of exposure to the new EE was sufficient to trigger a significant AIS length shortening (Fig. 4F, paired t -test $**P = 0.0035$). These results strengthen the evidence that homeostatic scaling is a continuous process when adapting to novel environments.

Minor comments

1) Presentation of spike traces in Figures 3 and 5 may be reversed for control and manipulated groups, because the order is opposite in other panels.

Thank you, this has been corrected.

2) In Figures S2 and S4, the first component of action potential phase plot did not differ between control and manipulated groups. Does this mean that factors other than sodium current, such as potassium current, were involved in the regulation of neuronal excitability?

Thank you for raising this point. We agree, this could have been clarified in more detail. The first peak in the AP phase plane plot has been shown to be influenced mainly by the onset position of the AIS (Hamada et al. 2016, Goethals and Brette et al. 2020). For this revision, we have reanalyzed our data for AIS onset location and found no significant difference between the groups (Fig. S3, Fig. S5). In fact, in most layer II/III pyramidal neurons, the AIS onset is directly adjacent to the border of the soma (see Fig. S3C, left panel). Thus, the finding that there is no change in the first component of the AP is consistent with our anatomical data and in support of the hypothesis that activity-dependent AIS plasticity primarily alters AIS length and thereby the current threshold.

3) Related to the above issue, it is better to explain how the increase of AIS length could reduce the threshold current.

We have included this point in the discussion (p. 19-20), highlighting that an increase in AIS length will increase the total number of sodium channels and thus sodium current, and is likely associated with a more distal AP initiation site. Both factors will reduce the rheobase current.

4) AIS length increased with long-term sensory deprivation (> fifteen days), but not with short-term one (five days). On the other hand, AIS length decreased within one hour. Why was the time course so different between the manipulations? It may also be interesting to discuss the functional implication of this variation.

Thank you for pointing this out. We have revised this aspect extensively in the discussion (p. 18-19), also in response to reviewer #2, point 3. We have expanded on potential molecular mechanisms, especially regarding

the hypothesis that internalization of AIS proteins is a more likely factor than actual disassembly and re-assembly upon rapid length increase, simply because this is a time-consuming task. At this point, however, we can only speculate about functional implications and therefore have kept this part to a minimum. We suggest that the AIS is a cellular feature that contributes to individual neurons maintaining an optimal physiological range of function when network states change.

5) Why was AIS apparently faint and wavy after treating slices with KCl or bicuculline? This may imply that the manipulations affected the organization of the entire AIS and/or the axon in addition to the AIS length.

The “waviness” is a phenomenon that we have intermittently observed in various preparations, including cryosections and acute slices, independent of treatment (see also Fig. 3D). We suspect it to be the result of variations in fixations which was not always present. Therefore, we have replaced the image with an example from another slice where the AIS appeared straight. The faintness however, although hard to reliably quantify, could indeed reflect biological changes after high KCl/Bicuculline treatment, e.g. protein endocytosis or proteolysis.

6) It was not explained why AIS length returned to baseline at three hours after the treatment of KCl or bicuculline.

The reviewer raises an interesting aspect for which we have no conclusive answer. The most parsimonious explanation is that the impact of elevated K^+ or bicuculline may be wearing off with time as a result of the fact that the AIS adapts (!). In both cases, this reduces the excitability of the local network within the slice preparations, reducing the increased excitation and therefore the possibility to, in the context of normal levels of excitability, re-adjust to the normal AIS lengths. This is in our view homeostasis.

Reviewer #2:

In their study, Jamann et al. reveal AIS plasticity in layer 2/3 of the mouse barrel cortex using two different in vivo manipulations: whisker deprivation as well as sensory enrichment. By doing so they demonstrate that the AIS can physically increase as well as decrease its length in a homeostatic manner under these conditions.

The authors first show the developmental time course of AIS length changes in layers 2/3 as well as 5. Overall, AISs seem to grow until around P10-P15, then shorten again a bit before stabilising. Interestingly, the time period of AIS shortening coincides with the onset of active exploration and whisking, raising the question whether this is an activity-dependent and not merely a genetically predetermined process. To address this, the authors next turned to whisker trimming in order to reduce sensory input. This resulted in longer AIS lengths, even in the adult cortex. Electrophysiological recordings confirmed that the physical AIS changes correlated with a functional decrease in current threshold. This strongly suggests that the AIS is undergoing homeostatic plasticity in response to reduced synaptic inputs caused by the manipulation, independent of developmental stage. A common criterion for establishing a plasticity as homeostatic is bidirectionality. Here, Jamann et al. are able to demonstrate that AISs in layer 2/3 of the barrel cortex are also able to undergo shortening by using enriched environment to increase sensory input. As before, this form of AIS plasticity was accompanied by a change in current threshold. Interestingly, unlike sensory deprivation, this manipulation resulted in rapid structural changes, with a significant change already observed after 1 hour instead of after 2 weeks, and an equally fast recovery of the AIS length back to baseline after only 6 hours. This rapid time course of AIS length changes was replicated in a final experiment in vitro.

Overall, this is an interesting and compelling study that probes AIS plasticity using in vivo manipulations, demonstrating that the AIS can increase as well as decrease its length in a homeostatic context. In particular,

the fast AIS shortening after sensory deprivation is of significant scientific impact, as it suggests, for the first time, that AIS plasticity is an ongoing process utilised during normal behaviour *in vivo*.

There are a few places where further clarifications will be beneficial, as outlined below. To convincingly support their findings, the authors should address the following, specific comments:

1) The short-term plasticity seen after sensory enrichment is particularly exciting, as it suggests that AIS plasticity is an ongoing, relatively dynamic process in vivo. In light of this it would be interesting to understand what happens at the 6 hour time-point, when AIS length returns to baseline. Does this happen because the enriched environment is simply no longer novel and thus the whisking activity has returned to normal? While returning animals to their home cage after 3 hours addresses this point to some extent, it would be helpful to know what happens if the animals were placed into a novel enriched environment after 3 hours.

The in vitro data doesn't sufficiently address this, especially without any measure of activity levels within the slices. These could have returned to normal after 3-6 hours, due to e.g. other homeostatic processes, slice deterioration, etc.

In addition, a thorough discussion of this point appears to be lacking. Do the authors suggest that the return to baseline is an intrinsic property of rapid AIS shortening, rather than activity-dependent?

The reviewer brings up an interesting point, which we addressed with new experiments. Here, mice (P28, n = 6) were subjected to the same EE conditions as before for 6 h, but then were exposed to a new EE for 1 h (new cage with different toys, but overall the same amount of stimulus). We hypothesized that - should the AIS indeed support homeostatic scaling *in vivo* - this new environment should elicit AIS changes just as rapidly as before, even in animals that had habituated and showed baseline AIS length after 6 h. Indeed, consistent with this idea, the AIS length was again significantly reduced in animals exposed to the 6 h + 1 h EE paradigm (paired t-test **P = 0.0035). These findings are included as Fig. 4F and in the text.

2) For this set of experiments, Jamann et al. have chosen to use as control the opposite hemisphere, after whisker trimming, presumably to reduce inter-animal variability. Particularly compelling is the finding that cfos+ neurons contralateral to the stimulated side show AIS plasticity, whereas cfos- neurons don't. The authors should consider adding data from ipsilateral (control) cfos+ and cfos- neurons.

Thank you for this suggestion. We have repeated the analysis for the contralateral hemisphere. However, only at the 3 h time point there were sufficient numbers of c-Fos⁺ neurons for a robust analysis. Indeed, the effect is reproducible on the contralateral hemisphere, with shorter AIS in activated c-Fos⁺ neurons (unpaired t-test **P = 0.0046). We have added the analysis to Fig. S4B.

3) The authors interpret their data as evidence that AIS lengthening only occurs over prolonged periods of time, whereas AIS shortening can be achieved within an hour. They suggest that this might be due to complex molecular interactions and energy-consuming protein assembly and transport. However, after AIS shortening they observe an equally rapid return of AIS length to baseline values, which suggests that the AIS can lengthen quickly, at least from a shortened state. It would be helpful if the authors could add this point to the discussion, together with possible reasons for this disconnect in plasticity phenotypes.

Thank you for this suggestion, which we followed by revising this aspect extensively in the discussion (p. 18-19), also in response to reviewer #1, point 4. We have expanded on potential molecular mechanisms, especially regarding the hypothesis that internalization of AIS proteins is a more likely factor than actual disassembly and re-assembly upon rapid length increase, simply because this is a time-consuming task. At this point, however, we can only speculate about functional implications and therefore have kept this part to a minimum. We suggest that the AIS is a cellular feature that contributes to individual neurons maintaining an optimal physiological range of function when network states change.

4) Finally, while the electrophysiological changes seen in this study are small, the correlation of AIS length with current threshold is very powerful. Since AIS plasticity can manifest as both a change in length as well as a change in position, it would be interesting to also see the correlation of this with current threshold.

We have reanalyzed the data of the neurons that were filled with biocytin, and measured the gap between the AIS onset and the soma. This analysis has been added to the supplements (Fig. S2, S4). However, we did not observe any significant correlation between the AIS onset and the current threshold. Of note, in most cases the AIS originated directly from the soma, and the spread in gap length is 0 to 2 μm , which might explain the lack of a role for AIS location in this cell type. The analysis indicates that the main factor that controls the current threshold *in vivo* is indeed AIS length and not location. We have added this point to the discussion (p. 19).

Minor points:

1) *Figure 1: It would be helpful if the authors could add an example AIS image for the E20 time point.*

We have added example images for E20 in Fig. 1A. However, since at that age, a clear distinction between layer II/III and V is not yet possible, we have opted to include images from directly below the pial surface (substitute for layer II/III) and adjacent to the future white matter (for layer V).

2) *On page 5, the authors state: "Bilateral whisker trimming resulted in a significant lengthening of AIS at all time points in Group 1 in layers II/III (...), but not in layer V (Fig S1A)."*

Figure S1A doesn't actually show all time points from Group 1, hence this statement is misleading.

Thank you for pointing this out. The statement has been corrected and this Fig is now Fig S2 due to the inclusion of a new Fig S1.

3) *Throughout the main text, synaptic currents are merely called PSCs, whereas it is clear from their methods that they are recording EPSCs rather than a mix of inhibitory and excitatory PSCs. This should be changed to remove ambiguity.*

We have clarified this section both in methods and the relevant figure legends. Since the recording voltage of -90 mV is more hyperpolarized than the chloride reversal potential of -70 mV in our conditions, the recorded spontaneous currents were presumed to be a mix of excitatory and inhibitory synaptic currents and therefore referred to as "PSCs".

4) *Figure 6: The figure legend states that 3-4 slices were used per condition, however there are clearly more than 4 points on the plots in both 6B and 6C.*

Thank you for pointing out this inconsistency. We always included minimally 2 animals per condition, thus the total number of data points can amount up to 8. We have specified this in the legend to make it clearer, double checked all values and corrected when necessarily.

5) *On page 16, the authors state: "After the onset of active whisking [...], a reduction in AIS length was observed, followed by a gradual length increase throughout adulthood." I cannot see any evidence for the quoted length increase. Please clarify.*

Thank you for pointing this out, it is indeed actually a slow decrease until adulthood. We have edited the sentence to reflect that after the onset of active whisking and explorative behavior at P12-13, a reduction in AIS length was observed, followed by a gradual length decrease until adulthood.

6) Page 25, methods, the authors state: "Staining was carried out as described above, however with prolonged incubation periods to allow for sufficient penetration of the tissue by the antibody". Please specify the length of this 'prolonged incubation period'.

We have specified the incubation periods in the methods (p. 26).

Reviewer #3:

The paper describes axon initial segment (AIS) plasticity in the somatosensory system using the relevant behavioural tasks. They show bidirectional AIS plasticity in this system and also test electrophysiological properties of neurons endured AIS structural plasticity, confirming physiological adaptation in response to the structural plasticity. They show this plasticity is restricted to the layer 2/3 neurons and not happening in layer 5 pyramidal neurons. Also, this study characterises developmental AIS plasticity in the barrel cortex and altogether add to the previously reported AIS plasticity in the visual and auditory systems. Experimenting of AIS in C-Fos activated neurons is a major advantage of this study confirming the plasticity as a result of changes in activity at single neuron level, basically linking in vitro data to the in vivo data in a practical way.

The findings are novel for the somatosensory system and in line with the previous reports in visual and auditory system with minor system-specific differences. The methodical and analytical approach are very clear and precise and covers a range of technical expertise, all well-executed.

A few minor points:

Minor points:

1) *It is not clear why β IV-spectrin has been used for AIS length measurement and AnK G for immunoblot analysis? Why not use one of them for both for consistency? Any technical reason?*

The reviewer correctly saw this putative discrepancy between immunoblotting and immunofluorescence. For immunoblots, we looked at ankG because it is the master scaffold protein and its isoforms have been associated with distinct roles at various neuronal sites. For length quantification, however, we mostly used β IV-spectrin because it was the more robust label for all *ex vivo* slices. To show the structural overlap of ankG and β IV-spectrin expression along AIS in pyramidal neurons, we have added new example staining with histogram traces from our analysis software in Fig. S1A.

2) *I am not convinced of the conclusion that 15 days of deprivation is necessary for AIS plasticity to happen, the only experiment for this conclusion is data obtained from group 2. What if the 10 days deprivation would be between 10-20 days? My argument is not this additional group to be done, as the study does a thorough check of the different windows already, but this part of conclusion should be toned down.*

Thank you for this valid point. Indeed, we never applied a 10-day deprivation window, and have therefore rephrased our statement to highlight the fact that other time windows may also elicit AIS plasticity (p. 18).

3) *In supplementary tables, there are typos in regard to RN and RS unit of measurements, (M Ω)*

Thank you, this has been corrected. Data from tables has been moved into the results section.

4) *Discussion in general is good and has addressed most of the findings but has missed on some of the literature on AIS plasticity, in particular the only reference available on the effects of enriched environment on AIS length (Nozari et al., Dev Psychobiol, 2017). It would be very interesting to see if the reported AIS plasticity in the primary somatosensory cortex can be communicated to the secondary sensory areas or association areas and at least the evidence can be discussed.*

We appreciate the suggestion to include Nozari et al's work, which we have now added to our text (p. 2). However, in light of an already comprehensive discussion and without any of our own data on secondary sensory cortex regions in our model, we respectfully decline to add further speculation to this certainly very interesting topic. Of note, the major body of our discussion covers the activity-dependent mechanisms of AIS plasticity, and the Nozari paper focuses on structural changes exclusively.

REVIEWERS' COMMENTS

Reviewer #1 (Remarks to the Author):

The authors have done a great job of addressing all of my concerns. I think the manuscript is entirely appropriate for publication in Nature Communications.

Reviewer #2 (Remarks to the Author):

The authors have sufficiently addressed my concerns. In particular, it is great to see they took the advice to add an extra experiment, showing renewed AIS plasticity in a second novel environment. This really adds to establishing AIS plasticity as a continuous process. I have no further reservations and recommend this interesting and compelling study to be published in Nature Communications.

Winnie Wefelmeyer

Reviewer #3 (Remarks to the Author):

The revisions in relation to my points are adequate. I have no further comments